

**Behaviour of Dissolved Phosphorus with the associated nutrients in relation to phytoplankton**
**biomass of the Rajang River-South China Sea continuum**
Edwin Sien Aun **Sia**[1], Jing **Zhang**[2], Shan **Jiang**[2], Zhuoyi **Zhu**[2], Gonzalo **Carrasco**[3], Faddrine Holt
**Jang**[1], Aazani **Mujahid**[4], Moritz **Müller**[1]
[1]Faculty of Computing, Engineering and Science, Swinburne University of Technology, Sarawak
Campus, Jalan Simpang Tiga, 93350, Kuching, Sarawak, Malaysia.
[2]State Key Laboratory of Estuarine and Coastal Research, East China Normal University, Zhongshan
N. Road 3663, Shanghai, 200062, China.
[3]Tropical Marine Science Institute, National University of Singapore, 119223, Singapore.
[4]Department of Aquatic Science, Faculty of Resource, Science and Technology, University Malaysia
Sarawak, 93400 Kota Samarahan, Sarawak, Malaysia.
Corresponding Author*: Moritz Müller, mmueller@swinburne.edu.my
**Abstract**
Nutrient loads carried by large rivers and discharged into the continental shelf and coastal waters are
vital to support primary production. Our knowledge of tropical river systems is still fragmented with
very few seasonal studies available for Southeast Asia for example, despite estimates that these
systems are among the hotspots globally for nutrient yields. The Rajang river, the longest river in
Malaysia, is a tropical peat-draining river which passes through peat-domes in the estuary and has
mass discharge of organic matter into the South China Sea. Three sampling campaigns (August 2016,
March 2017 and September 2017) were undertaken along ~300 km of the Rajang river to study both
spatial and seasonal distribution of nutrients and its fate in the coastal region. The analyses for
nutrients encompass both inorganic (i.e Nitrate, $NO_3^-$, Nitrite, $NO_2^-$, Ammonium, $NH_4^+$, Phosphate,
$PO_4^-$ (DIP)  and Silicate, dSi) as well as organic (Dissolved organic nitrate, DON and Dissolved
organic phosphate, DOP) fractions. It was found that DIP concentration was not seasonally influenced
but was spatially different along the salinity gradient whereas DOP was both seasonally and spatially
different. Both DIP and DOP exhibited non-conservative behaviour in the mixing. DIP was subjected
to 57.78% removal whereas DOP was subjected to 44.07% addition along the salinity gradient
towards the South China Sea. The bulk of the dissolved phosphate is from DOP (73.84%), in which
both DIP and DOP may have contributed to the phytoplankton biomass. Spearman's correlations
show that there was a switch in preference for DOP as compared to DIP depending on the
concentrations of DIP or DOP due to seasonality. The main limitation in the Rajang River was
assumed to be DIP based on the Redfield ratio. During the dry season, the $NO_3$-N:DIP ratios were



lower, which were ideal conditions for phytoplankton proliferation while in the wet season, the
increased $NO_3$-N:DIP ratios led to lower phytoplankton biomass. Overall, the Rajang River exports
0.12 t DIP mth$^{-1}$ into the South China Sea which is relatively low as compared to other major peat-
draining rivers in the world. At the current pace of deforestation and the projected intensification of
rainfall in the region, this finding provides an important baseline of the inventory of DIP into the
South China Sea. Our results also show that local variations are important to consider for future
models and that the assumption /generalization of SEA as a nutrient hotspot might not hold true for all
regions and requires further investigations.


Keywords: Dissolved inorganic phosphate, dissolved organic phosphate, Rajang River, South China
Sea, phosphate limitation




## 1.0 Introduction

The view of rivers as passive transporters have been recently been deemed null by studies (Richey et al., 2002; Tranvik et al., 2009). Aufdenkampe et al., (2011) and Marwick et al., (2015) states that rivers are now well acknowledged as key players in regional and global carbon budgets, with the majority of the fraction of terrestrial input are processed along the transit towards the coastal zone.

As the major pathway for nutrients dispersal from the continents to the oceans is through riverine transport (Liang and Xian, 2018), the N and P riverine loading to the estuarine ecosystems have increased on a global scale due to nutrient enrichment (Nixon, 1995). Nonetheless, eutrophication occurs due to enhanced nutrient levels vary from one aquatic environment to another (Di and Cameron, 2002). While tropical aquatic environments support an extensive amount of biodiversity, there are little to none studies of nutrient mass balances of tropical regions (Liljeström, Kummu and Varis 2012). Furthermore, Yule et al., (2010) and Smith et al., (2012) stated that tropical estuaries are the most biogeochemically active zones which are much more vulnerable towards anthropogenic nutrient loading as compared to estuaries at higher latitudes. Due to rapid economic development as a result of population growth, resulting in the extensive modification tropical South East Asian rivers and degradation of catchments (Jennerjahn et al., 2008; Yule et al., 2010). This is even more true for peat draining rivers which consequently includes the limited studies of nutrient transport and in particular the dynamics of phosphate (P) in such environments.

The Rajang River is subjected to human developments which may alter the quantity and quality of nutrients as well as the carbon (Rixen et al., 2016) and its influence on nutrient dynamics and the subsequent alterations towards primary productivity and microbiological function (Henson et al. 2018). Primary productivity and biomass accumulation in coastal and freshwater ecosystems are driven by seasonally high $NO_3^-$ concentrations (Kristiansen et al., 2001; Sieracki et al., 1993). However, as the Rajang river is tidal influenced, and consists of fluvially-driven inputs of terrestrial mineral soils in the upper altitudes and drains peat domes in the lower altitudes (towards the coastal regions), thus, it is imperative to understand the anthropogenic variability in nutrient dynamics in the landscape to better understand how such systems may respond to disturbance.

A macronutrient that is essential but often limiting in freshwater systems is phosphorus (Elser et al., 2007) and in under specific conditions also limit the primary productivity of terrestrial and coastal ecosystems (Street et al., 2018; Sylvan et al., 2006). In the second half of the 20th century, anthropogenic activities have caused the global riverine phosphorus and nitrogen inputs to increase by three times (Jennerjahn et al., 2004). On a global scale, it was estimated that the riverine DIP loading for the world's largest rivers which includes 37% of the earth's watershed area as well as half of the earth's population is 2.6 Tg yr$^{-1}$ (Turner et al., 2002). This value will undoubtedly increase due to the



increasing anthropogenic pressures. Runoff and leaching from animal production and agricultural
fields (Van Drecht et al., 2009) would lead to changes in primary productivity, ecosystem functioning,
hypoxic events, harmful algal blooms, damaged water quality as well as the increased greenhouse gas
emissions (Schindler, 1974; Deemer et al., 2016; Macdonald et al., 2016; Ho and Michalak, 2017).

The carbon pools in tropical peatlands are globally significant, with the current estimates ranging
from 40 to 90 Gt of C (Yu et al., 2010; Page et al., 2011; Warren et al., 2014). The disturbance of
peatlands due to anthropogenic activities such as deforestation and conversion of peatlands for
agricultural activities poses a threat to the environment. This is because disturbed peat soil changes
from carbon sink into carbon source, contributing to the greenhouse gases in the atmosphere (Hirano
et al., 2012; Hooijer et al., 2010). Recent studies of lateral transport of $CO_2$ of tropical peat-draining
rivers (Müller et al., 2015; Wit et al., 2015), the tropical peat-draining river of Maludam National Park
seem to have a moderate amount of outgassing of $CO_2$ as compared to other peat-draining rivers
globally. Globally, while the Rajang River is considered a medium-sized river based on its discharge
(Sa'adi et al., 2017), 11% of its catchment area is part of the 15-19% global carbon peat pool in South
East Asia (Page et al., 2011). Therefore, due to the knowledge gaps of tropical peat-draining rivers,
particularly the Rajang River, it is essential to understand the influence of peat on the riverine
phosphate loading into the South China Sea. As the South China Sea supports one third of the global
marine biodiversity (Ooi et al., 2013), the contribution of the Rajang River towards the South China
Sea in terms of primary productivity cannot be ignored.

Therefore, the aim of this study is to 1) better understand the spatial and temporal distribution of
nutrients, with particular focus on dissolved inorganic phosphate (DIP) and dissolved organic
phosphate (DOP) in the Rajang River with consideration to the diverse inputs and influences and 2)
consequentially determine its influence on the phytoplankton biomass.


**2.0 Methodology**

**2.1 Study Area**
The samples that were collected for nutrient analyses is as shown in **Fig. 1**. The red triangles
represent the samples collected from the dry season whereas the blue circles represent the samples
collected for the wet season.

The Rajang River is located in the state of Sarawak of Malaysia, which is located on the north-
western region of the Borneo Island. Based on the statistics provided by the Malaysian Department of
Statistics, (2019), the level of urbanization within the Sarawak state was at 53.8% of which the



estimated total population in Sarawak for the year of 2018 was 2.79 Million with a GDP of RM
113.982 billion in 2017. Two monsoonal periods occur within this region, whereby the southwestern
monsoon which occurs from May until September is normally associated with relatively drier weather
(hereafter referred to as the dry season) whereas the northeastern monsoon which is normally
associated with enhanced rainfall and subsequently frequent flooding occurs between the months of
December to February (herefore referred to as the wet season). Nonetheless, as put forth by Sa'adi et
al., (2017), rainfall is high throughout the year despite the monsoon which is associated with the drier
season. The discharge rates for the Rajang river drainage basin varies from $1000 - 6000$ m$^3$s$^{-1}$ for each
month (data obtained from 30 years of rainfall data) whereby the average is around 3600 m$^3$s$^{-1}$.
Rajang river drainage basin area is approximately 50,000 km$^2$ (Staub et al., 2000). Apart from that, the
proximal hills region also releases discharge and sediment whereby its delta plan covers
approximately 6500 km$^2$. Its delta plain contains low-ash, low-sulphur peat deposits which can be
greater than 1 m thick. According to Nachtergaele et al., (2009), 11% of the catchment size
corresponds to peatlands which extends over the aforementioned area. Furthermore, only 1.5% of
Sarawak's 17% of peatlands (out of 23% throughout the whole country) remains entirely pristine
(Wetlands International, 2010). In the upper reaches of the Rajang river, it drains mineral soils until
the town of Sibu, from which multiple distributary channels branch out and drains peat soils instead.

In this study, four distributaries (Igan, Paloh, Lassa and Rajang distributary) were studied. As put
forth by Staub et al., (2000), these extensive peatlands drain directly into the aforementioned
distributaries. Industrial oil palm plantations (Gaveau et al., 2016) as well as sago plantations
(Wetlands International, 2015) were converted from a majority of these peatlands, accounting for
more than 50% of the peatlands (11% of the total catchment size) in the Rajang watershed (Miettinen
et al., 2016). Timber processing, logging and fisheries are the main socioeconomic activities for the
local residents (Abdul Salam and Gopinath, 2006; Miettinen et al., 2016). According to (Müller-Dum
et al., 2019), saltwater intrusion occurs until a few kilometres downstream of the town of Sibu
whereas tidal influence extends further inland up to 120 km to the town of Kanowit (Staub and
Gastaldo, 2003) .

**2.2 Sampling**
The sampling area was divided into four categories according to salinity and source types: (1) marine,
(2) brackish peat, (3) freshwater peat, and (4) mineral soil based on the salinity profiles. The
classification of land-use is based on descriptions by Wetlands International, (2015), Gaveau et al.,
(2016), Miettinen et al., (2016) and Ling et al., (2017) to assess the possible anthropogenic influences.
The classification of land use was categorized as: 1) coastal zone 2) coastal zone with plantation
influence, 3) oil palm plantation 4) human settlements 5) secondary forests. Samples were collected
over a span of seven days for the first survey and four days on the second survey. The first survey was



constructed to obtain spatial coverage on a higher frequency with marine and freshwater end-members
in mind while sampling on the second survey was carried out on a lower frequency but with similar
spatial coverage and end-members. The first survey, in August 2016 was during the dry season while
the second survey in March 2017 was carried out during the wet season. The temperature, salinity,
dissolved oxygen (DO) and pH were measured *in-situ* utilizing an Aquaread®. For the two sampling
campaigns, all samples were collected within the upper 1 m (surface) using 1 L HDPE sampling
bottles that were pre-washed with 4% hydrochloric acid (HCl) via a pole-sampler to reduce
contamination from the surface of the boat and engine coolant waters (Zhang et al., 2015). All
samples analysed for nutrients were filtered through a 0.4 μm pore-size polycarbonate membrane
filters (Whatman) into 100 mL bottles that were pre-rinsed with the filtrate. About 100 mL of the
filtrate was collected in pre-acid washed polyethylene bottles. The samples were killed with 10 μL of
concentrated mercury chloride, $HgCl_2$, and kept in a cool, dark room before chemical analyses. For
phytoplankton pigments, the samples (250 – 1000 mL) were filtered through 0.7 μm pore-size GF/F
filters (Whatman) and carefully wrapped in aluminium foil before being immediately stored at -20 °C.
All samples that will be analysed for nutrients were filtered through a 0.4 μm pore-size polycarbonate
membrane filters (Whatman) into 100 mL bottles that were pre-rinsed with the filtrate. About 100 mL
of the filtrate was collected in pre-acid washed polyethylene bottles. These samples were then killed
with 10 μL of concentrated mercury chloride, $HgCl_2$ and kept in a cool, dark room before chemical
analyses. For chlorophyll *a*, the samples (250 – 1000 mL) were filtered through 0.7 μm pore-size
GF/F filters (Whatman) and carefully wrapped in aluminium foil before being immediately stored at -
20 °C.

**184    2.3 Nutrients Analyses**

The concentrations for nutrients were determined in the laboratory utilizing a Skalar SAN[plus] auto
analyser (Grasshoff et al., 1999). The components of nutrients that were measured include: Nitrate
($NO_3^-$), Nitrite ($NO_2^-$), Ammonium ($NH_4^+$), Dissolved Inorganic Phosphate (DIP), Dissolved Silicate
(dSi), Total Dissolved Nitrogen (TDN) and Total Dissolved Phosphate (TDP). The sum of $NO_3^-$, $NO_2^-$
and $NH_4^+$ were classified as dissolved inorganic nitrogen (DIN) whereas the concentrations of the
dissolved organic phosphorus (DOP) and dissolved organic nitrogen (DON) were calculated by
subtraction of DIP from TDP and DIN from TDN respectively via oxidation with potassium
persulfate digestion method (121°C, 30 min digestion) (Ebina et al., 1983). The component that was
not examined in this study is the exclusion of particulate P in the total determination of P loading.
While DIP is more biologically available as compared to particulate P (PP), Harrison et al., (2019)
suggested that Particulate P is usually the dominant form of P that is being exported to the coastal
areas. Thus, the bioavailability of particulate P should be further studied and modelled to better
understand the significance of P loading model outputs. However, as suggested by Jordan et al.,
(2008), most of the biologically available DIP in estuaries is converted from fluvial PP which is





enhanced by increasing salinities. Consequently, the DIP in estuaries could serve as a proxy for the PP
that originated from headwaters and its importance can still be reflected in the concentration of
biologically available DIP. The analytical precision for all nutrients components measured was <5%.
In order to analyse correlation between humic acids and DIP or DOP, dissolved organic carbon
concentrations (DOC) were used as a proxy as part of the hydrophobic fraction of dissolved organic
matter are generally derived from humic substances (Findlay et al., 2003). Lastly, for DOC
concentrations the results were obtained from Martin et al., (2018) whereas SPM values were reported
by Müller-Dum et al., (2019).

**2.4 Chlorophyll a determination**
As a proxy for phytoplankton biomass, chlorophyll a (Chl *a*) was utilized. The extraction of Chl *a* is
as provided by (Martin et al., 2018). The filters were grounded with methanol and extracted with an
ultrasonicator (VCX644, Sonics and Materials, USA) in an ice bath. Then, 0.45 µm PTFE membrane
was utilized to filter supernatant of the extracts after centrifugation at 3,000 rpm. For the analyses of
pigments, a HPLC system (Agilent 1100 series) was used based on the methodology of Zapata et al.,
(2000) and Zapata and Garrido, (1991). Chl *a* standards were purchased from Sigma-Aldrich.

**2.5 Data analyses**
The spatial distribution of the physico-chemical parameters were plotted in Surfer 13.and all graphs
were plotted utilizing GraphPad. Averages of measured parameters were reported as ± Standard Error
(SE) unless stated otherwise. For statistical correlations, SPSS (IBM SPSS Statistics 22) was utilized
for calculations of Independent sampling *t*-test (between seasons), one-way ANOVA (between source
types) and Spearman's ranking (Bivariate correlation, for nutrients correlation). Graphs were
produced using Prism 6 (GraphPad Software, Inc).


**2.6 Export calculations**
For calculations of the discharge of the entire Rajang river, precipitation values were obtained for the
entire Rajang river catchment which was obtained Tropical Rainfall Measuring Mission (TRMM)
website (NASA, 2019). The precipitation values were converted into m$^3$ from mm and multiplied by
the conversion factor to obtain the discharge s$^{-1}$ and further multiplied with 60% (0.6) (Whitmore,
1984) to obtain the discharge values after taking into consideration the surface run-off values.
Furthermore, the value for the entire catchment area was derived from the values provided in Müller-
Dum et al., (2019).

$Discharge = Mean\ precipitation \times area\ of\ basin \times conversion\ factor\ to\ s^{-1}$
$\times surface\ runoff\ percentage$






237 River loads for DIP and Si were calculated for the entire Rajang river with the assumption that the

238 total loading from the headwaters from the Upper Rajang river (input) would equal to the output (into

239 the South China Sea). The freshwater end-member concentrations of DIP were obtained based on the

240 average concentrations ($\mu$mol L$^{-1}$) of based on the nutrient concentrations of the samples obtained at

241 salinity $\approx$ 0 (Liang and Xian, 2018). The average concentrations were then used for the estimation of

242 river loads utilizing the equation provided in Müller-Dum et al., (2019) with slight modifications

243 provided by the conversion factor from (ICES, 2019).

244

245 The nutrient loads of Phosphate Phosphorus (PO$_4$-P) were obtained from DIP and were calculated

246 based on the conversion factors (ICES, 2019) whereby:

247

248 $$1\ \mu g\ PO4\ L^{-1} = 1\ \div MW\ PO_4\ \mu g\ L^{-1} = 0.010529\ \mu mol\ L^{-1}\ = C$$

249 $$C = \text{conversion factor for DIP}$$

250 $$f = \text{conversion factor from s}^{-1} - \text{y}^{-1}$$

251 $$g = \text{conversion factor from g to t}$$

252 $$d = \text{discharge (m}^3\ \text{s}^{-1})$$

253 Hence, the equation for yield is as stated below:

254 $$t\ DIP\ mth^{-1} = Conc.\ of\ Average\ DIP\ \times C\ \times Discharge\ \times f \div g$$

255

256

257

258 **3.0 RESULTS**

259 **3.1 Physico-chemical parameters and nutrient concentrations**

260 The physico-chemical parameters of temperature (°C), salinity (PSU), dissolved oxygen, DO (mg L$^1$)

261 and suspended particulate matter, SPM (mg L$^{-1}$) of dry and wet seasons were plotted along the Rajang

262 River-South China Sea continuum (**Fig. 2**).

263 Based on **Supp. Table 1**, the temperature in the dry season was 29.92 $\pm$ 0.20 °C whereas for the wet

264 season the temperature was 28.54 $\pm$ 0.30 °C. For both seasons, the variation of temperature between

265 the cruises was limited (**Fig. 3.2**). The full range of salinities freshwaters to marine waters were

266 covered in both cruises, ranging from 0 to 33 PSU. In the dry season, dissolved oxygen ranged

267 between 2.7 mg L$^{-1}$ to 4.9 mg L$^{-1}$ whereas in the wet season, the range was from 4.5 – 7.58 mg L$^{-1}$.

268 The mean values for dissolved oxygen increased by nearly two-folds during the wet season with an

269 average of 6.03 $\pm$ 0.17 mg L$^{-1}$ as compared to the dry season with an average of only 3.84 $\pm$ 0.11 mg

270 L$^{-1}$. The SPM concentrations of both the dry and wet seasons decreased from headwaters (freshwater





mineral soil) towards the coastal region (marine) with a range of $25.01 - 161.27$ mg L$^{-1}$ in the dry
season and $36.06 - 494.46$ mg L$^{-1}$ in the wet season.
The nutrient concentrations of dissolved inorganic nitrate, DIN (µM), dissolved organic carbon, DOC
(mM) and dissolved silicate, dSi (µM) were plotted in **Fig. 3** as shown below.
The range of DIN in both dry and wet seasons is from 7.1 to 28.7 µM. However, the measured DIN
concentrations for the dry season varied, with the highest mean occurring in the brackish peat $21.86 \pm$
$1.59$ µM as compared to marine, freshwater peat and freshwater mineral soils ($11.36 \pm 1.69$ µM ,
$13.33 \pm 1.14$ µM and $10.90 \pm 1.76$ µM, respectively). In terms of DOC, the concentrations ranged
from 0.08 to 0.40 µM (Martin et al., 2018). For dSi, the range in the dry and wet season was from $4 -$
$179.1$. The dSi concentration in the wet season had an average of $147.72 \pm 32.79$ µM as compared to
the dry season with an average $106.67 \pm 11.06$ µM. The concentrations of dissolved inorganic
phosphate, DIP (µM), dissolved organic phosphate, DOP (µM) and total dissolved phosphate, TDP
(µM) were plotted as shown in **Fig. 4**.

From **Fig. 4**, the range of DIP is from $0 - 0.27$ µM.  The overall range of DOP for both seasons is
from 0.04 to 0.11 µM.  Combining the two parameters (DIP and DOP), the concentrations of TDP
generally increased with mean concentrations ranging from $0.23 - 0.42$ µM during the dry season and
$0.16 - 0.42$ µM during the wet season. Collectively, the range of TDP is from $0.13 - 0.53$ µM 0.13 to
0.53 across both seasons.

DIP ranged from $0 - 0.27$ µM (**Fig. 5**). The overall range of DOP for both seasons was between
0.04 and 0.11 µM. Combining the two parameters (DIP and DOP), the concentrations of TDP
generally increased with mean concentrations ranging from $0.23 - 0.42$ µM during the dry season and
$0.16 - 0.42$ µM during the wet season. Collectively, the range of TDP is from $0.13 - 0.53$ µM across
both seasons. The concentrations of DIP and DOP were also plotted along the integrated conservative
mixing line against salinity (**Fig. 5(A and B)**). In terms of the DIP concentrations, both dry and wet
season consistently increased from headwaters towards the coastal region with the mean
concentrations of each source type ranging from $0.03 - 0.17$ µM whereas the wet season had mean
concentrations of $0.06 - 0.13$ µM. On the other hand, DOP concentrations during the dry season were
relatively stable with a mean concentration of $0.23 \pm 0.01$ µM. In contrast, the mean concentrations
during the wet season increased from headwaters towards the coastal region ($0.09 - 0.33$ µM). The
total DIP in dry season represents 26.16% of the total TDP pool whereas the DOP in dry season
represents 73.84% (TDP represents 100%) (**Fig. 5(C)**). On the other hand, DIP pools in the wet
season represents 34.70% of the total TDP pool whereas DOP represents 65.30% of the total TDP
pool. The average concentrations for DIP when they are classified under different land use are
$0.11 \pm 0.02$ (coastal zone), $0.117 \pm 0.019$ (coastal zone with plantation influence), $0.087 \pm 0.012$ (oil





palm plantation), 0.085± 0.027 (human settlement) and 0.032 ± 0.031 (secondary forest), respectively
(Fig. 3.5(D)). In terms of dSi, based on **Fig. 5(E)** and Table 2, it was found to be negatively
correlated to both dry and wet seasons (-0.819 and -0.550, respectively) whereby the dSi:DIP ratios
drastically decreased along the salinity gradient. Lastly, there were no significant correlations between
DIP as well as SPM in both dry and wet seasons.  However, when plotted against salinity, it was
shown that the SPM:DIP ratios were varied in the wet season and increased along the salinity gradient
in the dry season (**Fig. 5(F)**).

**3.2 Nutrient Ratios across the Rajang River-South China Sea continuum**
The DIN:DIP ratios were high throughout the Rajang River (**Table 1**), which can be correlated with
the low DIP concentrations. The same trend can be seen for the other two nutrient ratios (Si:DIP and
Si:DIN). In a study carried out by Liang and Xian, (2018),the two components that were utilized were
the $NO_3$-N:DIP as these two were the main components that were utilized or incorporated by
phytoplankton for growth. Hence, for discussion in this study, the $NO_3$-N:DIP were utilized for
discussions.
Based on Table **2**, the parameters which were highly positively or negatively correlated with DIP in
the dry seasons were DON, Silicate, Salinity and DO (-0.520, -0.819, 0.839 and -0.537, respectively)
whereas for DOP in the dry season, none of the parameters were highly correlated. On the other hand,
in the wet season, the parameters that were highly correlated with DIP were DON and Silicate (-0.631
and -0.550 respectively) whereas for DOP, the parameters that were highly correlated were DOC, dSi
SPM and Salinity (-0.688, -0.557, -0.844 and 0.880 respectively).
**3.3 Factors influencing phytoplankton biomass**
DOP was further plotted against DOC (**Fig. 6(A))** against the salinity gradient in which there is an
observed trend whereby there is an increase in DOP with the decrease in DOC concentrations along
the salinity gradient. From **Table 3**, the parameters that were positively correlated with Chl *a* in the
dry season were DIP and TDP (0.562 and 0.631, respectively) and negatively correlated with dSi (-
0.796). In the wet season, Chl *a* was found to be positively correlated with DOP, TDP, Salinity (0.692,
0.770 and 0.815, respectively) and negatively correlated with dSi and SPM (-0.713 and -0.733,
respectively). Chl *a* was plotted against salinity and compared with the dSi as well as SPM (**Fig.
6(B and C)**; **Table 3**) and showed that Chl *a*:dSi ratios increased significantly only in the dry
season. For SPM, while SPM decreased drastically in the wet season and remained fairly constant in
the dry season, the Chl *a*:SPM ratio was found to increase along the salinity gradient only in the dry
season.



### 3.4 P yield calculations and comparisons with other global peat-draining rivers

Among the tropical/subtropical blackwater rivers compared (**Table 4, Fig. 7**), the highest yields based on Fig.6 was the Amazon River (377.39 t DIP y$^{-1}$) followed by the Pearl River (29.30 t DIP y$^{-1}$). Next, the Siak River had DIP yields of 21.63 t DIP y$^{-1}$. The Rajang River and the Dumai River have yields of 1.41 t DIP y$^{-1}$ and 0.001 t DIP y$^{-1}$, respectively.

### 4.0 Discussion

### 4.1 DIP sources and behavior

The concentrations of DIP increased from the headwaters from mineral soils to the coastal region along with salinity ($F_{(3, 40)}= 12.009$, $\rho = 0.000$ (**Fig. 4** and **Table 1**). However, the difference in DIP concentrations between the dry and the wet season was not found to be significant ($t_{(42)}=-0.514$, $\rho = 0.610$). The increase in DIP towards the coastal region can be supported by Froelich et al., (1985) and Fox, (1990) which showed that there may be probable desorption of DIP from particles as well as estuarine and marine sediments (Caraco et al., 1990; Pagnotta et al., 1989) that was caused by increasing salinities (Zhang and Huang, 2011).

Non-conservative behaviour was observed in the dry season (**Fig. 5(A)**), indicating a constant removal of DIP towards the coastal region (average of 57.87% removal across both seasons, **Supp. Table 2**). This was similar to DIP behaviour shown in the Changjiang estuary (Kwon et al., 2018) which showed possible $PO_4^{2-}$ removal within the estuary due to biological removal or buffering actions of suspensions and sediments of the estuary, the phosphate buffering mechanism. Furthermore, studies in Europe and North America (Lebo and Sharp, 1992; Nixon et al., 1996; Sanders et al., 1997) also show large scale removal of DIP by suspended particles in estuaries. In the wet season, DIP showed non-conservative behavior as well. The varying DIP concentrations might indicate probable point sources of DIP. In another study by Ling et al., (2017) on the Rajang river, it was reported that the total phosphorus and SRP (DIP) was higher in the stations located at the upper part of river. However, this study was carried out only during the wet season and in tributaries different to this study. Hence, the values obtained could likely originate from point sources. Another possible explanation for the increase in DIP is due to the resuspension of sediments as shown by the higher SPM levels (**Fig. 2**) near the coastal region. Oenema and Roest, (1998) stated that the bioavailability of P transported from land is only a fraction whereby its movement is determinant on the transport and mobilisation of soil particles (Jarvie et al., 1998; Stanley and Doyle, 2002). Furthermore, as put forth by Stumm and Morgan, (1996), 10% of naturally weathered phosphorus are





only available to the marine biota in the form of orthophosphate (i.e. DIP). As shown in **Fig. 5(D)**, it

is likely that the concentration of dissolved inorganic phosphate originated from probable leaching

from anthropogenic activities (from oil palm plantations) as well as desorption from sediments under

increasing salinity (coastal zone). It is interesting to note that in a study by Funakawa et al., (1996) on

peat soils in Sarawak, the concentrations of N and P were fairly high in the soil solution, even in those

classified as oligotrophic peat, except for the concentrations of P adjacent to the centre of the peat

dome. However, depletion of phosphate was observed during the rainy season at a sago plantation

farm grown on deep peat which was associated with the clear-cutting of forests and the successive

disruption in nutrient cycling. Thus, it can be inferred that the higher average DIP values in the wet

season (**Fig. 5 (C)**) as compared to the dry season in this study was a result of probable run-off from

the disturbed peat.

### 4.2 DOP sources and behaviour

With relation to the TDP (**Fig. 5(C)**), the DOP represents a significant percentage compared to the

DIP pool. Even though there is mounting evidence that phytoplankton and/or zooplankton and even

microbial populations are able to hydrolyze a considerable amount of DOP in natural waters (Chrost

et al., 1986), many studies exclude DOP and it is hence infrequently measured. It is, however, of

importance to consider DOP when assessing nutrient budgets and nutrient limitations (Monbet et al.,

2009). It was shown that DOP (referred to as Filtrate Hydrolysable Phosphate) formed 85% of the

Total Filterable Pool (Ellwood and Whitton, 2007) with DOP originating from the drainage of peat

and underlying limestones. Both dry and wet seasons showed addition of DOP (44.07% addition, see

**Supp. Table 2**) towards the coastal region (**Fig. 5(B)**). Based on the independent t-test, DOP

differed slightly between dry and wet seasons ($t(22.218)=1.777$, $\rho = 0.09$) but was significantly

different between source types ($F(3,41)=3.927$, $\rho = 0.015$). Furthermore, DOP concentrations were

negatively correlated with DOC (-0.688, as shown in **Table 2** and **Fig. 6(A)**) in the wet season

which was in line with a study by Whitton and Neal, (2011) who showed that DOC concentrations

were low when the DOP pools were at its highest. Besides probable sources such as sewage effluents

or agricultural soils, Whitton and Neal, (2011) also showed that DOP pools in downstream sites might

have originated upstream but have yet to be utilized by organisms or be hydrolysed by soluble

phosphatases in the water. In the wet season, the concentrations of DOP exceeded that of the dry

season (**Fig. 6(A)**), likely due to the higher run-off induced by higher precipitation during the

sampling campaign. According to Nissenbaum, (1979), it was estimated that 20-50% of the organic

phosphorus reservoir in sediments are bound by humic acids. As a large proportion of peat is made up

of humic substances (Klavins and Purmalis, 2013), the draining of peat would then lead to the

probable release of high amounts of DOP. However, the highest correlation of humic substances

(DOC) was with DOP during the wet season (-0.688, see **Table 2**). A similar pattern was observed for

DOC run-off from the peatlands (Martin et al., 2018) which was accelerated by higher precipitation as



indicated in the steeper DOC gradient in the wet season in **Fig. 6(A)**, suggesting probable higher
DOP run-off as compared to DOC. This was in line with a prediction model by (Harrison et al., 2005)
in which DOC:DOP ratios tend to be lower in regions with intensive agricultural activities.

### 4.3 Nutrient ratios and fate in the estuarine and coastal region

Generally, the ratios for $NO_3N$:DIP are extremely high (**Table 1**), indicating that the river is naturally
low in phosphate which could possibly be limiting nutrient in the Rajang river. According to Justić et
al., (1995), P limitation could potentially occur when N:P is greater than 22. Based on the $NO_3N$:DIP
ratios in the dry season, the ratio of 17.74 (1.15), is less than the aforementioned possible P limitation
(when N:P>22) as suggested by Justić et al., (1995). Hence, the dry season is in favour of the
Redfield's ratio of 16:1, indicating optimal conditions for phytoplankton growth as compared to the
wet season. Si limitation occurs when Si:DIN is greater than 1 and Si:P is less than 10. In the Rajang
River, the Si:P ratios were higher than the Redfield ratio across both seasons and source type. All Si:N
ratios were higher across both seasons and source type except for the dry season ($0.42 \pm 0.04$, **Table**
**1**). Cloern, (2001) and Kemp et al., (2009) highlighted that estuaries that are highly turbid, strongly
mixed and exchanged high amounts of organic inputs from the livestock production or watershed with
agricultural activities will not exhibit a relationship between primary productivity and nitrogen.
However, in this study, the $NO_3N$:DIP ratios differed between the dry and wet seasons, especially
within the brackish peat region (**Table 1**). The $NO_3N$:DIP ratios were higher in the dry season as
compared to the wet season. This could be due to the increased DIN concentrations in the dry season
due to the decomposition of dissolved organic nitrogen as demonstrated by Jiang et al., (2019).
Furthermore, as shown in **Fig. 2**, the lower SPM levels in the brackish peat during the dry season
led to the enhancement of light which favours the growth of phytoplankton, which can be reflected in
the increased Chl *a* concentrations (**Fig. 6(B) and Fig. 3.6(C)**). The uptake of DIP by phytoplankton
may have led to the drawdown of DIP (Li et al., 2017). In estuarine zones, silicate is usually
conservative whereby it is influenced mainly by the flux from dry to wet season (Zhang, 1996). The
highly negative correlation between silicate (-0.796) and the positive correlation of DIP (0.562) in the
dry season with Chl *a* may explain the net removal of Silicate within the estuarine to coastal region by
phytoplankton i.e. diatoms and is enhanced by the increased presence of DIP. Conversely, in the wet
season, the intensity of ammonification and nitrification in the Rajang River was reduced during the
wet season, which led to lower DIN concentrations as compared to the wet season (Jiang et al., 2019),
thus reflecting the generally lower $NO_3N$:DIP ratios which were closer to but still not at the optimal
Redfield ratio. Furthermore, Chl *a* was not correlated with DIP in the wet season (**Table 3**) as
reflected in the higher $NO_3N$:DIP ratios (**Table 1**) in the brackish peat region in the wet season. This
was identical to the scenario in the Chesapeake Bay where phytoplankton bloom was delayed due to



higher rapid flushing in the wet season (Malone et al., 1988). When river flow was higher, the
downstream mass transport of biomass was relatively more important versus production utilizing DIP
as a source of biomass. In addition to that, during periods of high discharge (i.e. wet season), seaward
advective transport driven by freshwater inflow prevents biomass accumulation due to its flow being
faster than phytoplankton growth rate (Cloern et al., 2014). This can be further supported by the fact
that there was almost a two-fold increase in SPM (**Fig. 2**) during the wet season which could have
constrained phytoplankton production due to light attenuation and altered spectral quality sediments
(Wetsteyn and Kromkamp, 1994). Furthermore, during the wet season, the ratios for $NO_3N$:DIP were
much lower than in the dry season (**Table 1**), with the exception of the marine region which was
possibly caused by higher run-off of phosphates or nitrogen from anthropogenic activities such as oil
palm and sago plantation (**Fig. 5(D)**). As put forth by Tarmizi and Mohd, (2006), oil palm
plantations require more phosphate rock fertilizer in the mixing of the Nitrogen (N):Phosphate
(P):Potassium (K) ratios in order to compensate for the phosphates that are immobilized by the soils,
implying that there is an abundance of phosphates within the agricultural soils. This would support the
notion that greater run-off from higher precipitation during the wet season would lead to higher
leaching of phosphates into the Rajang river. While Thevenot et al., (2010) illustrated that tropical
soils are naturally poor in N and P compounds, intensive land-use changes such as deforestation will
increase recalcitrant compounds which are readily decomposed). Furthermore, drained peatlands
export more phosphorus than mineral soils after clear-cutting of peat forests as peat has lower
phosphate adsorption capacity (Cuttle, 1983; Nieminen and Jarva, 1996).
Numerous studies have shown the importance of DOP as a source of phosphorus (Bentzen et al.,
1992; Boyer, Joseph N.; Dailey et al. 2006)in aquatic environments to support algal metabolism and
growth when the bioavailable P pools drop below critical threshold concentrations with regards to
other requisite nutrients (Lin et al., 2016). It is more advantageous for phytoplankton to utilize DIP as
it can be directly taken up and assimilated; whereas, DOP, on the other hand, requires more energy
(Falkowski and Raven, 2013) as it requires phosphatases catalysing the hydrolysis of phosphate
monoesters found within DOP compounds. Consequently, this would result in the liberation of
inorganic phosphate as well as organic matter (Labry et al., 2005). Thus, as the Rajang River has a
greater pool of DOP as compared to DIP (**Fig. 5(C)**), it is evident that there is a probable switch in
preference for DOP as compared to DIP depending on the concentrations of DIP or DOP. From **Table
3**, the change of Chl *a* being positively correlated to DIP to DOP reflects a switch in the roles of DIP
and DOP as the preferred phosphate sources for the phytoplankton biomass. As further described by
Lin et al. (2016), the operational measurement of DOP is defined as the difference between TDP and
DIP, thus polyphosphate esters and inorganic polyphosphate as well as two other DIP species, which
are phosphite ($PO_3^{3-}$) and phosphine ($PH_3$), are included operationally in the determination of DOP.
This is reflected in the prediction of functional genes as shown in another study in **Supp. Fig. 1** which



indicate the presence of phosphonate and phosphinate metabolism in microbial communities
(including cyanobacteria) even though in low abundance.

### 4.4 Nutrient loads & Comparisons with worldwide systems: other peat and non-peat draining rivers

It should be noted that this paper discusses the estimation of P loads based on the freshwater inputs,
which excludes addition and removal (fluxes) from the calculations. As reported by Statham, (2012),
while freshwater inputs in estuarine environments will frequently be exceeded by tidally driven fluxes
of seawater, nutrients in river waters will typically have greater concentrations as compared to the
adjoining seawater. While the estimated figures in t P y$^{-1}$ (**Fig. 7**) are an underestimation due to the
exclusion of particulate phosphates and sedimentary phosphates, they are still useful for estimation
purposes.

Globally, while it was predicted by Seitzinger et al., (2005) that the river basins in both Central
America and South East Asia (Malaysia and Indonesia) are hot spots (within the top 10% globally) for
nutrient yields of various P forms), the export of P from Rajang is comparatively minor when
compared to other major rivers. This can be justified by Seitzinger et al., (2005), whereby the major
driver that controls export of P and P forms based on the model is water discharge. When compared
with other peat draining rivers in Southeast Asia, the Rajang river exports 1,178 times more t DIP y$^{-1}$
compared to the Dumai river, which is a pristine peat-draining river, whereas it was 15 times lower
than the Siak river (highly polluted blackwater river). When compared to the Amazon, the export of
the Rajang river was 267 times lower. Considering another major anthropogenically influenced river
draining into the South China Sea, the Pearl River (third largest river in China; Strokal et al., 2015),
the Rajang exports about 23 times less than the Pearl River. Comparing the dSi:DIP ratios to the
yields in the Rajang, showed that while DIP yields were variable, their sources are likely
anthropogenic in nature as dSi originates from natural chemical and physical weathering which are
relatively stable compared to riverine N and P loads (Beusen et al., 2009). In the Siak River, the
DIP:dSi ratios were the highest, however the yield of the Siak was lower than the Pearl as well as the
Amazon River. The yield of the Siak River was comparative with the Pearl River even though the
discharge for the Siak River was less was due to the domestic wastewater discharges which increased
the DIP concentrations by 470%. A similar pattern was observed in the Dumai River as well. While
the DIP yields of the Amazon as well as the Pearl River were higher than that of the Rajang River, the
DIP:dSi ratios were similar, indicating that the DIP yield In the Rajang River was likely
anthropogenic in nature. The vast difference in DIP yields in the Pearl River was due to agriculture
and industrial activities as well as sewage (Vitousek et al., 2009; Qu and Kroeze, 2012; Maimaitiming




et al., 2013). On the other hand, the DIP yield in the Amazon was the highest but was attributed to the
high discharge which was about 18 times higher than the Pearl River (Table 4). Even though the
addition as well as removal rate of both DIP and DOP is known, the P accumulation rate which is
largely dependent on several factors such as the sedimentation rate, bottom-water oxygen content is
largely unknown. By referencing the soil P:Si ratios (obtained from Funakawa et al., 1996) in a peat
swamp forest along the Rajang River, it can be inferred that the Rajang River may be subjected to
high burial and sedimentation of P, as reflected by the low DIP:dSi in the water column compared to
the soil. Since these estimations are only based on DIP exports, the actual P load of the Rajang River
and its contribution to the adjacent South China Sea and global P loads should be determined to better
inform government authorities for proper management of the Rajang river basin. As proposed by
Jiang et al., (2019), the mild DIN input likely supports primary productivity within the region.
Likewise, the P loads similarly contribute towards sustaining primary productivity and subsequently
the fisheries industry (Ikhwanuddin et al., 2011).


**5.0 Conclusion**

This study represents an in-depth look into the nutrient dynamics of the Rajang river and its
tributaries. The DIP concentrations in the Rajang River were variable with source types which
increased along the salinity gradient but were not significantly different between seasons. Seasonality
slightly exhibited for DOP but was significantly different between source types. Both DIP and DOP
exhibited non-conservative behaviour, with DIP subjected to 57.78% removal whereas DOP was
subjected to 44.07% addition along the salinity gradient towards the South China Sea. In the Rajang
River, the bulk of the dissolved phosphate is from DOP (73.84%), in which both DIP and DOP may
have contributed to the phytoplankton biomass. Spearman's correlations show that there was a switch
in preference for DOP as compared to DIP depending on the concentrations of DIP or DOP due to
seasonality.  The complexity of DOP formation, supply and degradation is due to the heterogeneity
which originates from variable as well as various origins such as river supplies, algal excretion, cell
lysis etc. as well as the degradation process of DOP (both enzymatic and chemical) is largely
unknown, which requires further examination.  During the dry season, the $NO_3N$:DIP ratios were
lower, which were ideal conditions for phytoplankton proliferation, while in the wet season, the
increased $NO_3N$:DIP ratios led to lower phytoplankton biomass. In terms of export loads of P, while
the Rajang River exports more DIP compared to Dumai (a pristine peat draining river), it is much less
compared to the Pearl and the Amazon river. In order to further understand the dynamics of
phosphorus on the Rajang River and the coastal region, long term observations with higher frequency
should be carried out. While the loading of P and is not as extensive as other major rivers, including
those that discharge into the South China Sea, with further understanding of the addition and removal
rates of the P components as well as the sedimentation rates, more can be known about the
contributions of P export from the Rajang River into the South China Sea which is essential as a
reference to improve regional as well as global P budget estimations.

**6.0 Acknowledgements**

The authors would like to thank the Sarawak Forestry Department and Sarawak Biodiversity Centre
for the permission to conduct collaborative research in Sarawak waters under the permit
NPW.907.4.4(Jld.14)-161, Park Permit No WL83/2017, and SBC-RA-0097-MM. Special mention to
the boatmen, in particular Lukas Chin while sampling along the Rajang River. We would also like to
thank Jin Jie for aiding with the nutrients analyses, Patrick Martin for providing DOC measurements
and Denise Müller-Dum for providing SPM measurements. The authors would also like to thank the
student helpers from UNIMAS, SKLEC, NOCS and Swinburne Sarawak who greatly assisted with
fieldwork and logistics. M.M. acknowledges funding through Newton-Ungku Omar Fund
(NE/P020283/1), MOHE FRGS 15 Grant (FRGS/1/2015/WAB08/SWIN/02/1) and SKLEC Open
Research Fund (SKLEC-KF201610).



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





**Tables**
**Table 1:** Nutrient ratios of the selected parameters along four source types (mean ± SE)

| Nutrients Ratios | Season | Source Type (Mean ± SE) | | | |
|---|---|---|---|---|---|
| | | **Marine** | **Brackish Peat** | **Freshwater Peat** | **Mineral Soil** |
| **DIN:DIP** | Dry | 73.61 ± 12.55 (n=3) | 203.36 ± 24.69 (n=13) | 404.50 ± 62.45 (n=4) | 438.00 ± 83.11 (n=8) |
| | Wet | 77.73* (n=1) | 152.78 ±19.01 (n=8) | 265.60 ±97.69 (n=5) | 161.81* (n=1) |
| **NO$_3$-N:DIP** | Dry | 17.74 ± 1.15 (n=3) | 114.63 ± 16.35 (n=13) | 209.19 ± 31.74 (n=4) | 229.39 ± 40.63 (n=8) |
| | Wet | 29.93* (n=1) | 69.85 ± 11.78 (n=8) | 199.49 ± 104.28 (n=5) | 112.87*(n=1) |
| **Si:DIP** | Dry | 31.86 ± 8.23 (n=3) | 883.04 ± 206.16 (n=13) | 4793.68 ± 923.36 (n=4) | 6615.26 ± 1429.10 (n=8) |
| | Wet | 119.57* (n=1) | 897.00 ± 182.63 (n=8) | 4001.02 ± 2183.14 (n=5) | 2458* (n=1) |
| **Si:DIN** | Dry | 0.42 ± 0.04 (n=3) | 3.90 ± 0.81 (n=13) | 11.71 ± 0.85 (n=4) | 16.47 ± 1.71 |
| | Wet | 1.04 ± 0.50 (n=2) | 5.40 ± 0.69 (n=8) | 12.10 ± 2.12 (n=5) | 15.19* (n=1) |

866                                                                    * Indicates only one sample






**Table 2:** Spearman's rank of various parameters against DIP and DOP in the dry and wet season.
Bolded values indicates greater significance with statistical significance (>±0.5)

| Parameters | Dry | | Wet | |
|---|---|---|---|---|
| | **DIP** | **DOP** | **DIP** | **DOP** |
| DIP | N/A | 0.237 | N/A | 0.416 |
| DOP | 0.237 | N/A | 0.416 | N/A |
| DIN | 0.476** | 0.005 | 0.447 | -0.282 |
| DON | **-0.520**** | -0.226 | **-0.631*** | -0.427 |
| TDN | -0.081 | -0.148 | 0.111 | -0.466 |
| DOC | 0.192 | 0.123 | -0.563 | **-0.688**** |
| dSi | **-0.819**** | -0.328 | **-0.550*** | **-0.844**** |
| SPM | 0.21 | 0.004 | -0.014 | **-0.557*** |
| Sal | **0.839**** | 0.453* | 0.450 | **0.880**** |
| DO | **-0.537**** | -0.121 | -0.207 | 0.413 |

870                                                                       ** means significant at the 0.01 level (two tailed)

871                                                                       * means significant at the 0.05 level (two tailed)


**Table 3**: Spearman's Rank of Chl *a* in dry vs wet with selected parameters. Bolded values indicates
greater significance with statistical significance (>±0.5)

| Season | Dry | Wet |
|---|---|---|
| **Parameters** | **Chlorophyll a** | |
| **DIP** | **0.562*** | 0.189 |
| **DOP** | 0.486 | **0.691*** |
| **TDP** | **0.631*** | **0.770*** |
| **Sal** | 0.618 | **0.815**** |
| **DIN** | 0.275 | -0.223 |
| **dSi** | **-0.796**** | **-0.713**** |
| **SPM** | -0.016 | **-0.733*** |
| **DON** | -0.291 | -0.499 |
| **DOC** | -0.209 | 0.545 |

876                                                                       ** means significant at the 0.01 level (two tailed)

877                                                                       * means significant at the 0.05 level (two tailed






**Table 4**: Comparison of nutrient concentrations of major global rivers or other peat-draining rivers *vs.*
Rajang river (μmol L$^{-1}$)

| River | Country | Catchment Size (km²) | Discharge (m3 s⁻¹) | Classification | DIP (μmol L⁻¹) | DOP (μmol L⁻¹) | dSi (μmol L⁻¹) | DIN (μmol L⁻¹) | Reference |
|---|---|---|---|---|---|---|---|---|---|
| **Pearl River** | China | 453,700 | 10,464 | Peat | 0.43 – 1.44 | 0.58 | 138.3 | 112.6 | Li et al., (2017) |
| **Rajang** | Malaysia | 52,009 | 3600 | Peat (11% of total) | 0.002 – 0.26 | 0.14 – 0.32 | 4.01 – 179.00 | 7.10 – 28.68 | This study |
| **Amazon (Morth)** | Brazil | 6,300,000 | 180,000 | Peat | 0.7 | - | 144 | - | Demaster and Pope, (1996) |
| **Dumai, Sumatra (Black water)** | Indonesia | 7,500 | 16 | Peat | 0.017 – 0.033 | - | 0.7 | 1 | Alkhatib et al., (2007) |
| **Siak, Sumatra (Polluted Black water)** | Indonesia | 10,500 | 99 - 684 | Peat (21.9) | 0.2 - 36.7 | - | 1.6 – 89.1 | 7.9 - 67.9 | Baum et al., (2007) |







**Figure Captions**

**Fig. 1**: Location of the Rajang River in Sarawak, Malaysia (Inset). Close up map of the Rajang basin and the stations sampled along the Rajang river and its tributaries (Red triangle: Dry season, Blue circle: Wet season). The bold cross indicates the location of Sibu.

**Fig. 2:** Distribution of temperature (°C), salinity (PSU), dissolved oxygen, DO (mg L$^1$) and suspended particulate matter, SPM (mg L$^{-1}$) in the dry and wet season along the Rajang River-South China Sea continuum

**Fig. 3:** Concentration of DIN (μM), DOC (μM) and dSi (μM) in both dry and wet seasons along the Rajang River-South China Sea continuum

**Fig. 4:** The distribution of DIP (μM), DOP (μM) and TDP (μM) concentrations in the dry and wet season along the Rajang River-South China Sea continuum

**Fig. 5: (A)** Distribution of DIP along salinity gradient in the dry and wet season and theoretical conservative line calculated based on integration. **(B)** Distribution of DOP along salinity gradient in the dry and wet season and theoretical conservative line. **(C)** Composition (%) of Phosphates in the Rajang River. **(D)** DIP composition based on different classifications/anthropogenic source **(E)** Ratio of dSi:DIP against salinity (PSU) **(F)** DIP:SPM  against Salinity (PSU) of surface waters along the Rajang River

**Fig. 6:** (A) Dissolved organic phosphate, DOP (μM) and dissolved organic carbon, DOC in both wet and dry season (μM) against salinity (PSU) (B) Chl *a*:dSi in dry and wet season against salinity (PSU) (C) Chl *a*:SPM in both dry and wet season against salinity

**Fig. 7:** The yield of DIP and the DIP:dSi ratio in selected blackwater rivers along increasing discharge (t DIP y$^{-1}$). The dotted line represents the DIP:Si soil reference for the Rajang River (Funakawa et al. 1996)





**Figures**

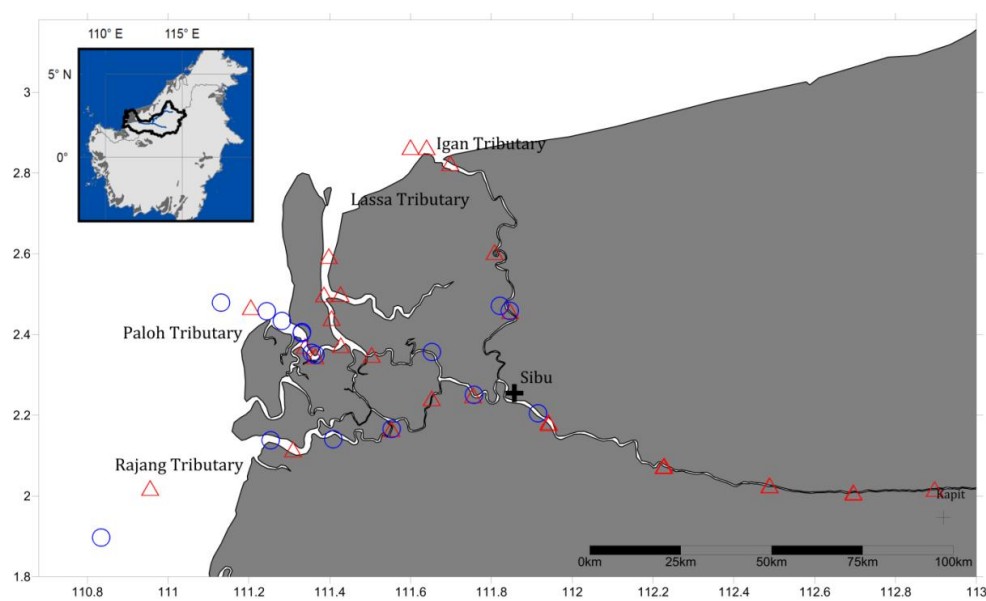

**Fig. 1**





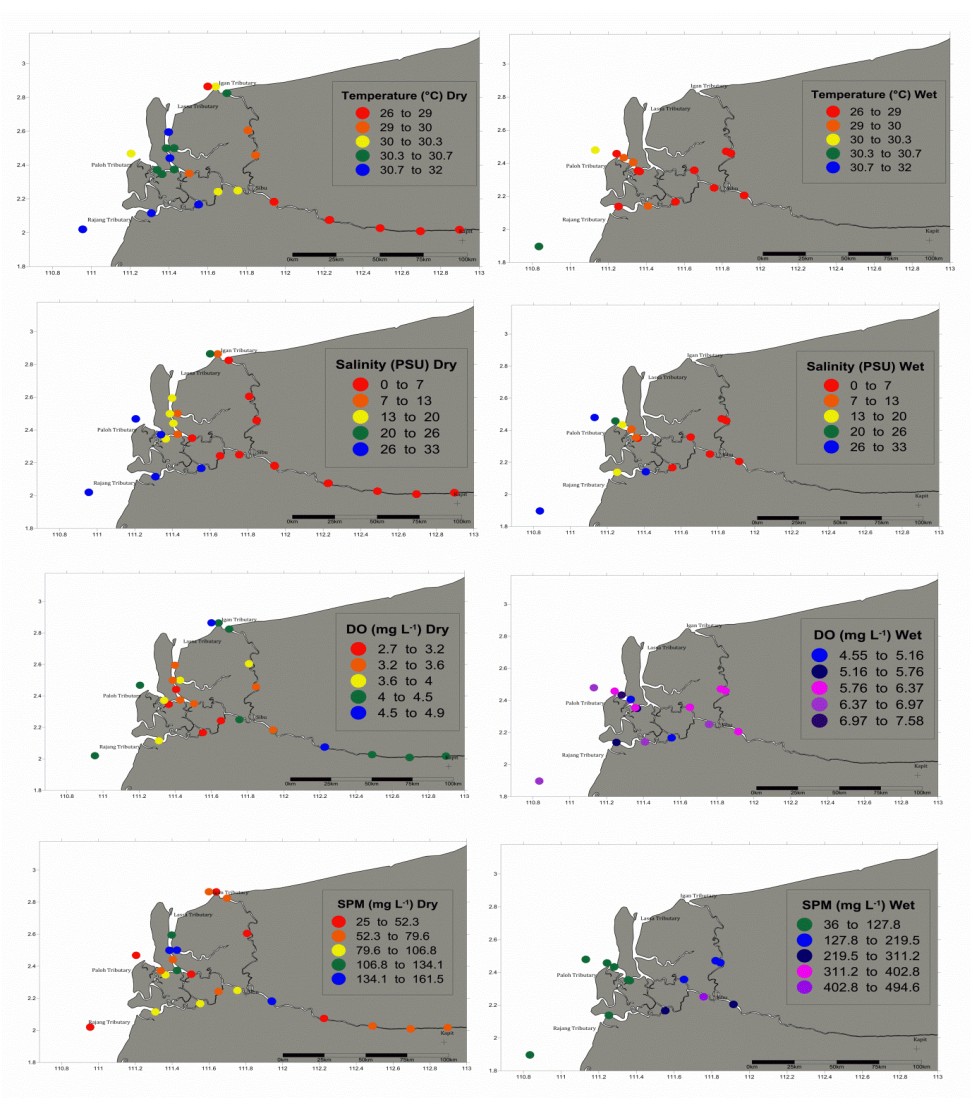


**Fig. 2**




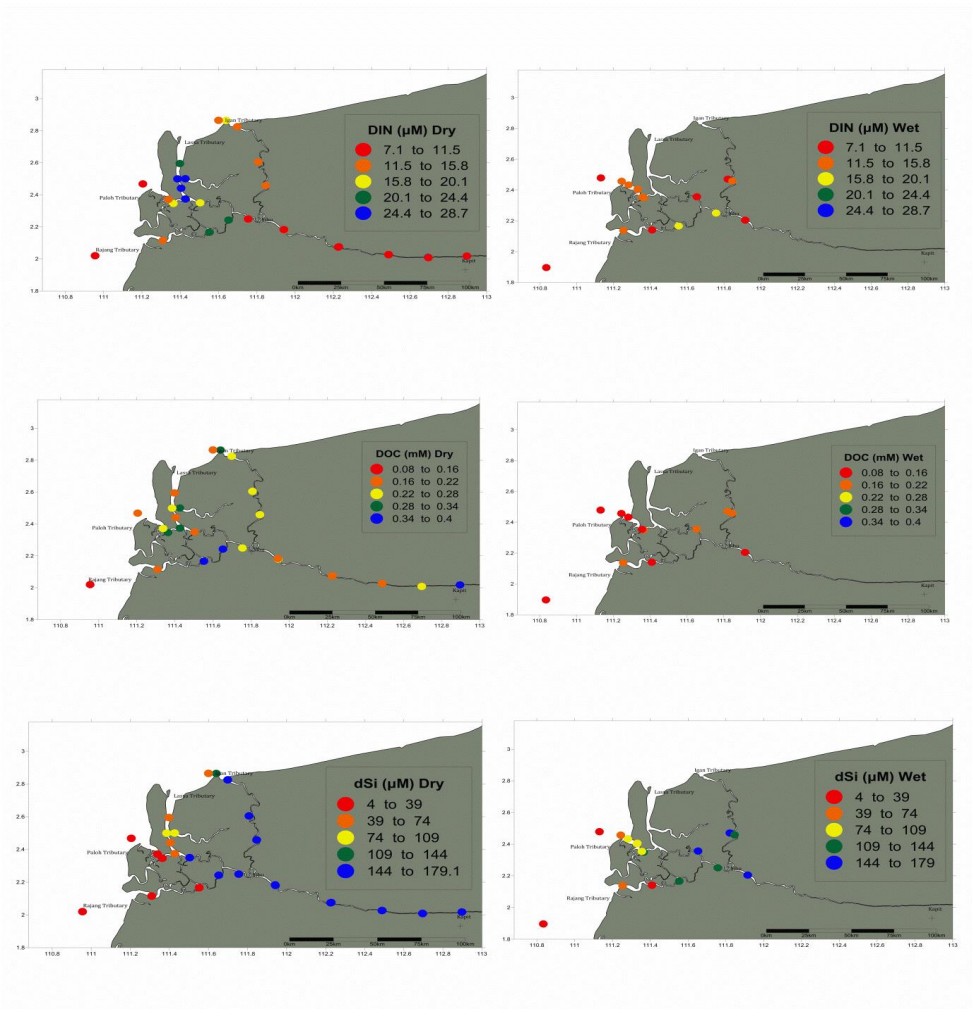


**Fig. 3**



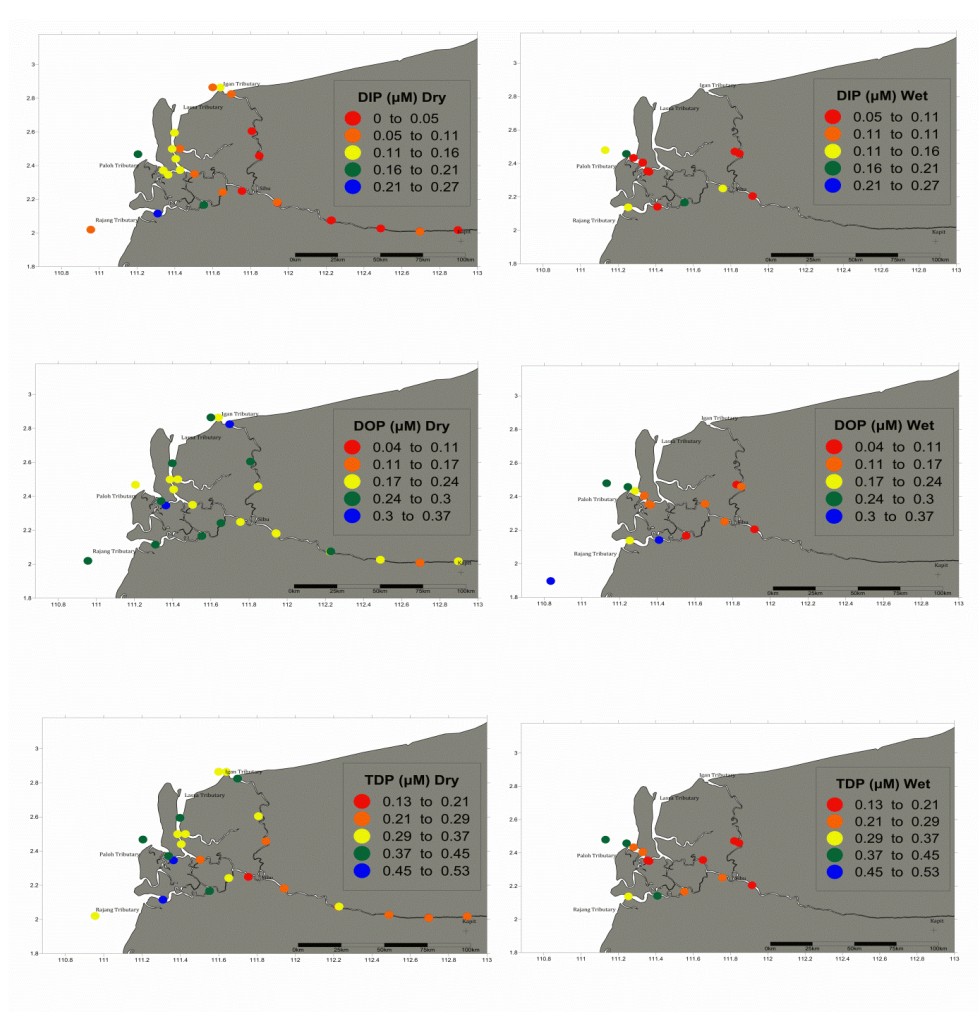


**Fig. 4**





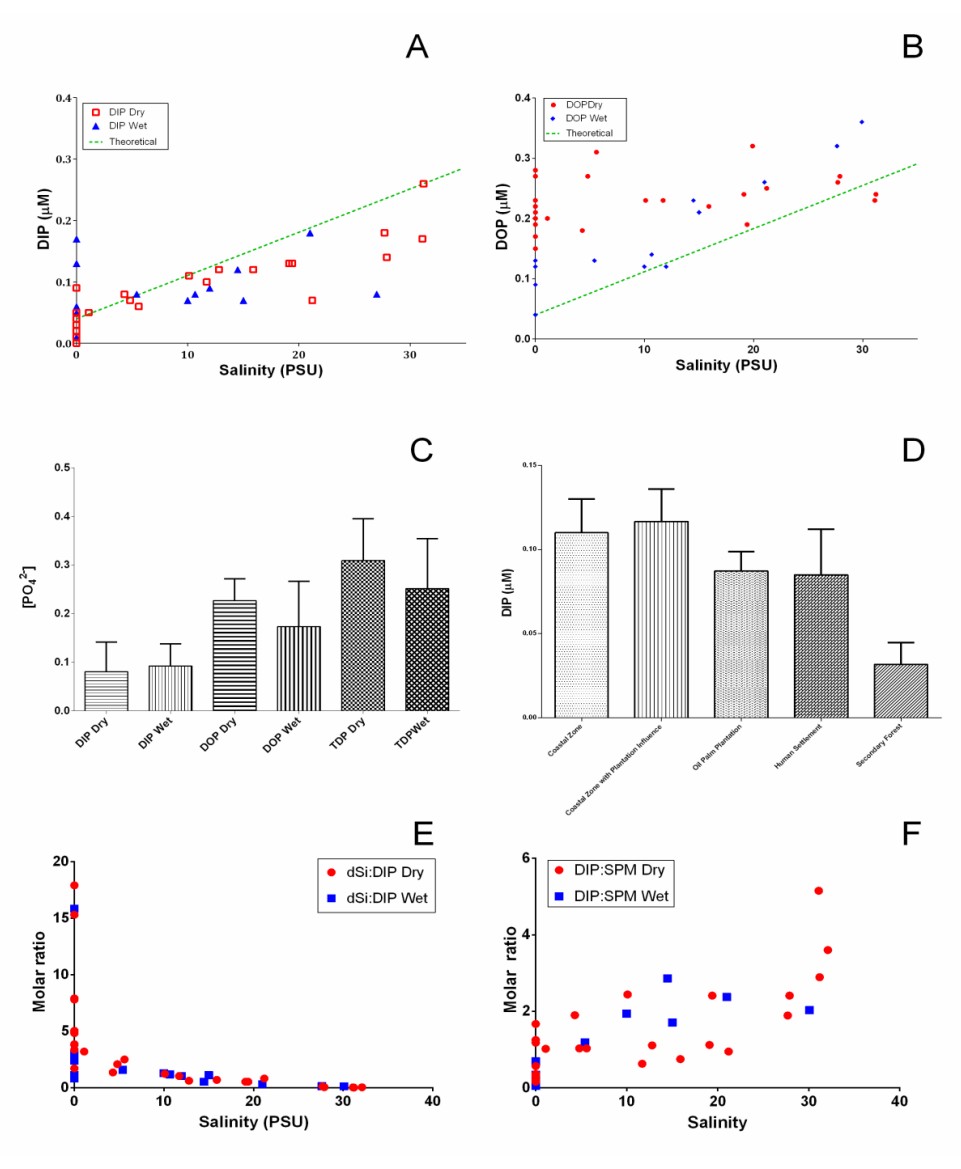


**Fig. 5**



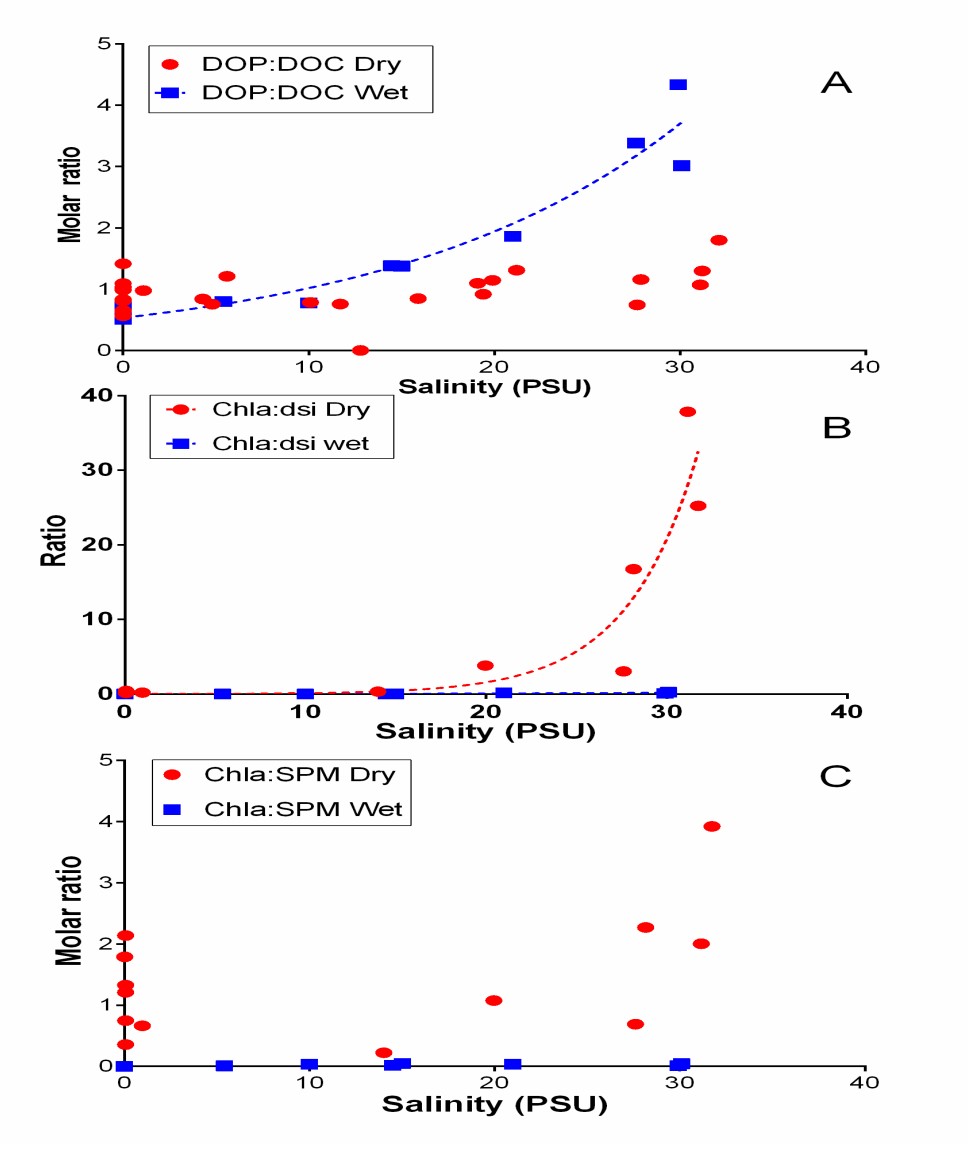


**Fig. 6**



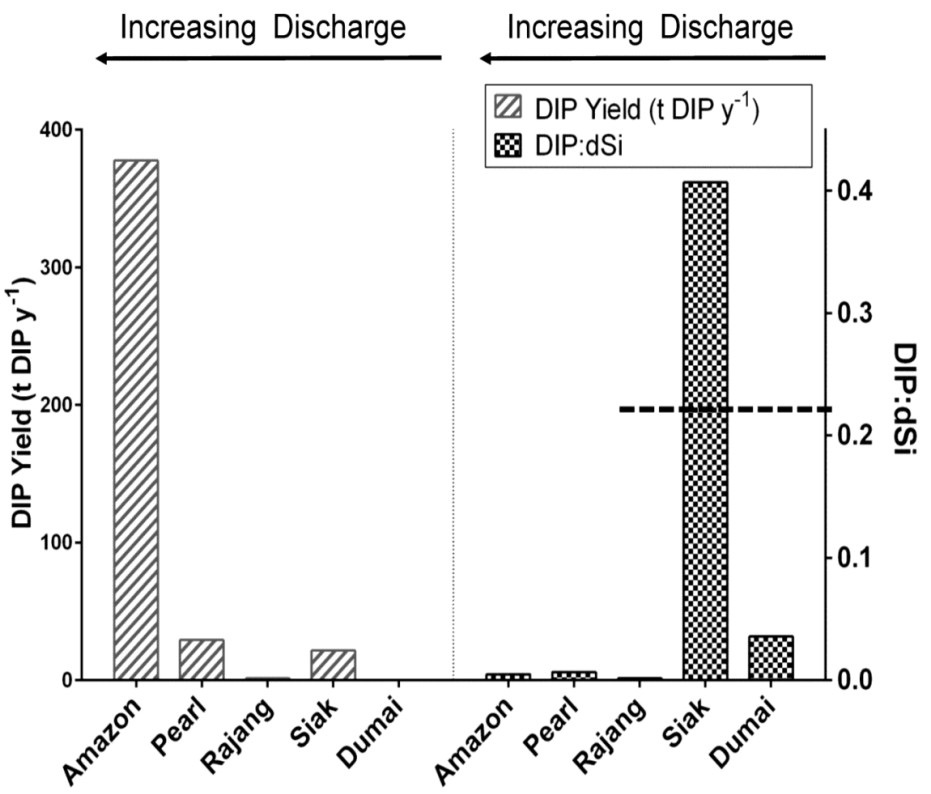


**Fig. 7**


