# Peer review of "Behaviour of Dissolved Phosphorus with the associated nutrients in relation to phytoplankton"

_Biogeosciences, 2019_

## Referee Comment (RC1) · Anonymous Referee #1 · 23 Sep 2019

General comments

The manuscript describes the results of water quality sampling procedures for the Rajang River -South China Sea continuum. The manuscript represents a contribution to scientific progress, presenting new spatially and seasonally varied data for the area of interest. The scientific methods and assumptions are clearly outlined, the estimation of DIP export to the sea is useful for understanding the system behavior. However, the major comments are related to a better definition of goals of the study, data visualization and interpretation of the results. In my opinion, the data collected and analyzed,

especially nutrient concentrations could be represented better, so that another type of comparison could be applied focusing more on measurements (e.g. faceted boxplots). For example, it is hard to identify the type of the sediment or the time of the sampling in Figures 2-4 and relate them to Table 1. In addition, I would focus more of the actual measurements, rather than ratios, or report both (Table 1, Figures 2-4). I would be more careful with statements about phytoplankton preferences of DIP and DOP based on correlations of these variables, in addition to low nutrient ratios leading to lower phytoplankton biomass. Furthermore, I think that conclusion about a particular nutrient limitation based on the ratios might be misleading, unless there are clear indications of low nutrient concentrations. The ratio can be high, however the concentrations of bioavailable nutrients could be also high, thus none of the nutrients might be limiting phytoplankton growth.

Specific comments

1. 19-21 Rework the sentence "..for example, despite. . ." 2. 25 "distribution fate?" 3. 25-27 Place abbreviations of water quality samples in parenthesis 4. 30-32 It is hard to understand what is "removal" means. Did DIP decreased by 57.78%? 5. 32-33 Suggest rephrase, not clear: The bulk = major fraction of? 6. 33-35 Which preference is it? 7. 36-38 Back to general comments: what if the increased $NO_3$:DIP ratio in wet season was due to higher discharge and consequently loading of $NO_3$? How can ratios lead to anything? 8. 65-67 Rephrase the sentence 9. 76-79 Too convoluted sentence, suggest divide into 2 10. 81-83 Too convoluted, suggest divide into 2 11. 85-87 Rephrase, too wordy 12. 93-112 Talking about Carbon here, but that is not the focus of this study. Basically there is a need in a smoother transition between the gaps in knowledge and the goals of the study 13. 118-120 Should go to figure caption. And similar paragraphs just occupy space and 14. 136-137 Change to "which can be thicker than 1 m" 15. 155-160 All these categories and classifications should be visualized on a study map (Fig. 1) 16. 164-165 So in Methods it is stated that there were 2 surveys, while in Abstract 3 sampling campaigns (Line 23) are mentioned. How many were

there? 17. 194-201 Looks more like discussion 18. 227 Change to "obtained from" 19. 248-254 Very confusing way of writing the equation. Why not state the equation, number it and explain the conversions, variables and units in the text? 20. 260-262, 273-274 Again, should be in a figure caption, or removed. It is a strange way to start a paragraph/section 21. 291 Starts with the same information as in 285. This should be cleaned 22. 316-317 DIN:DIP would be definitely correlated with DIP, because there is DIP on both sides 23. 320-321 Change or remove this sentence 24. 322-327 Which parameters are discussed? Was there any parameterization? 25. 350-351 DIP increases towards the sea while 358-359 says that there is a removal of DIP towards the coast. I am confused 26. 374 Use "is" instead of "are" 27. 378-385 It is hard to understand the connection between the citation and the idea. I see that Funukawa et al 1996 stated that N and P are fairly high in soil solution, but how from this sentence 383-383 can be concluded? 28. 394-395 Instead of "addition" it is better to use "increase" 29. 411-412 How can DOP and DOC be compared? 30. 416-420 Back to general comments: the ratio can be high but the concentrations also can be high 31. 443-444 Chl a can be uncorrelated with DIP, but how is this reflected by NO3:DIP ratios? 32. 447 Change "mass transport of biomass" 33. 473 Why use "Thus"? It is not clearly following from the previous sentence 34. 473-477 Is it really evident? I agree that DOP can be possibly utilized by phytoplankton, but the increase of DOP concentration does not indicate a preference switch. It is actually supported by discussion at 469-470 that DIP is easier to consume. 35. 491-493 Still did not understand why the estimated figures are useful 36. 495-498 Too convoluted 37. 498-499 It is unclear what exactly Seitzinger et al 2005 justifies 38. 505-508 Too convoluted, suggest split into at least 2 sentences 39. 532-555 Needs additional work as Conclusion is largely based on the points mentioned above

---

## Referee Comment (RC2) · Anonymous Referee #2 · 23 Sep 2019

I would like to thank the authors for their obvious hard work on this manuscript. The role of phosphorus in driving primary productivity is a focus in many systems (Lake Erie, Lake Taihu, Gulf of Mexico, etc.) and so to understand how this relationship behaves in as many different systems as possible is fundamental to eventually being able to design control and remediation protocols. I am glad to see a study that looks at the different fractions of dissolved phosphorus (DIP v. DOP), a subject that has been difficult to address in the past, but has been gaining in research focus recently. Additionally, I am pleased to see their focus on the role of the river itself as a fundamental actor in

this relationship as opposed to an inert transporter of nutrients from one place to the next. The role of in-stream processing on nutrient loading is poorly understood, and by showing that there are real differences in nutrient concentrations along the entire length of the river helps to show that rivers are chemically dynamic systems. I believe that this study helps to progress the state of the science, and should be considered for publication after some changes, particularly to the grammar and sentence structure of the manuscript.

Below are general comments about the manuscript, as well as more specific comments broken down by section.

General: As noted above, there are some issues with the language of the manuscript that makes it difficult to understand what the authors were trying to say. This become a problem in the discussion and conclusion sections where it seems the authors are contradicting themselves from one sentence to the next. I don't believe it is a misunderstanding or misinterpretation of their results, rather an issue with word choice and sentence structure. I think the comparisons with other rivers is a good idea, but ultimately executed poorly, it feels rushed and not properly fleshed out. I suggest that this section is a good starting point for another manuscript, but probably doesn't belong here.

Introduction: This section, in particular, will require editing/ rewriting. While the general structure of the section is fine, there are a significant number of grammatical issues which make reading and comprehension difficult. I have no issue with the message the authors are trying to convey; they did a fine job of providing supporting sources, however, it took several re-reads to be able to understand what they were trying to establish. Below are some examples of the confusing language used in this section, but is not a comprehensive list; these should be used as examples of what was outlined above.

Line 52: Awkward phrasing, try something like "The view of rivers as simply passive

transporters of nutrients has been challenged in a number of recent studies (Richey et al., Tranvik et al.)

Line 59-60: Confusing wording- why nonetheless? The previous sentence sets up the fact that eutrophication is increasing.

Line 65-67: Sentence fragment. I think the authors are saying "The rapid increase in economic development, driven by population growth, has resulted in the modification of SE Asian rivers and the degradation of their catchments."

There are numerous sentences like theses throughout the introduction, and they make the manuscript difficult to follow. The authors make some good points, and set up their study, it just takes a significant amount of effort to parse the language. This section has the ability to be a fine introduction if and when the language is corrected.

Methods: Study area: This section is fine, and the authors do a good job of describing their sampling locations/ decisions in selecting their sites. There are still some oddities in the language, but is ultimately easier to read and understand.

Sampling: Again, this section is generally fine, and does a good job of describing their sampling protocol, although I would ask how many samples were collected at each site as well as time of day for each collection. Are these single grab samples or are the authors averaging over a larger number of samples at each site? I may have missed it, but I did not see anything that describes this directly.

Nutrient Analysis: I am not sold on the use of DIP as a proxy for PP, particularly in areas away from the estuaries, but I don't think it would have a significant impact on this study's results.

Line 203: What fraction is it?

CHL-a determination: The methods used are fine, although for blue-greens, chlorophyll can be misleading, and perhaps phycocyanin would be a better measure.

[Figure]

Data analysis: The methods outlined and statistical software used are fine.

Export Calculations: I am not familiar with some of the calculations that they used here, but after looking into them I don't see anything that would raise any issues based on how they have described using them.

248-254: Is this the equation standard for this journal? Just seems like an odd way to write all of this out.

Results: There are many of the same sort of language issues in this section that were present in the introduction. While it doesn't nullify the results it does make it difficult to read in a timely way. The results section is excessively wordy, and feels like it was written in several different pieces and then combined instead of being a singular effort.

Lines 206-262: This seems like it should be a figure caption. There are a couple of other spots in this section with the same sort of "disconnected" feel. If you can use figures or graphs, do so, and limit the amount of writing, particularly in a results section

Line 316 and elsewhere: Be careful in how you describe your DIN:DIP ratio comparisons.

Discussion: I think that the language issues that came up in the introduction and results are present in this section as well. In a number of places it is not readily apparent what the authors are trying to say, and it takes multiple re-readings to understand. Additionally, there are a few places where they seem to contradict their own discussion points, but I think that it is through the use of incorrect phrasing as opposed to a misunderstanding of the results.

351-359: This is an example of what was described above, is it increasing or decreasing as it moves towards the coast?

378-381: What? Consider removing.

416-418: Ratios are not concentrations are not loads. Flow weighting the loads could

be helpful.

485-529: I think that this is an important topic, but feels "jammed-in" here, and doesn't really advance the narrative in the way I think the authors wanted- if anything it muddies things up a bit. I would cut this section way, way down or remove altogether. It is the seed of another manuscript to be honest and is not done justice here.

Conclusion: Again, I think language issues hinder the author's ability of bring a significant amount of work to a fine enough point. The authors are trying to extend their results into places I'm not sure they actually go. This study is a good survey of the P exports of the river, and describes spatial and temporal variability in those measurements, but it is dangerous to compare to other systems (i.e. The Detroit River exports significantly more N and P to Lake Erie than the Maumee River, but the Maumee has an outsized role in harmful algal bloom formation due to the concentrations of those nutrients, and its relatively warmer water).

Tables and Figures: Table formatting is odd. This may be due to the way it printed out for me, but there are line jumps and returns that should be removed.

Figure 2-4: the dots are difficult to see when printed out- mainly there is not enough contrast between the points and the map base layer.

Figure 5 and others: Be careful with axis font sizes, they are all over the place and make it difficult to read a number of the plots.

Figure 6 and 7: Look weirdly stretched out, like they were not resized properly.

---

## Author Comment (AC1) · 4 Nov 2019

General comments The manuscript describes the results of water quality sampling procedures for the Rajang River - South China Sea continuum. The manuscript represents a contribution to scientific progress, presenting new spatially and seasonally varied data for the area of interest. The scientific methods and assumptions are clearly outlined, the estimation of DIP export to the sea is useful for understanding the system behavior. R: We would like express our gratitude to Ref #1. The comments and suggestions provided helped to improve the manuscript significantly.

However, the major comments are related to a better definition of goals of the study, data visualization and interpretation of the results. R: Noted and improved as outlined below.

In my opinion, the data collected and analyzed, especially nutrient concentrations could be represented better, so that another type of comparison could be applied focusing more on measurements (e.g. faceted boxplots). R.: Thank you for highlighting this. For Figures 2-4, we chose to display nutrient concentrations on a map to ease the discussion of the spatial patterns. For Figures 3-6, the ratios help us to illustrate the importance of specific components. The theoretical line for both DIP and DOP concentrations were used to reflect conservative mixing and enable a discussion on the addition or removal of DIP/DOP. We do, however, agree that boxplots of the nutrient concentrations would be useful too and will add them in the supplement.

For example, it is hard to identify the type of the sediment or the time of the sampling in Figures 2-4 and relate them to Table 1. R: We do agree and will add the figures in the supplement to ease the discussion.

In addition, I would focus more of the actual measurements, rather than ratios, or report both (Table 1, Figures 2-4). R: Thanks for the suggestion. All values are now reported in Table 1. As mentioned above, new figures will be added in the supplementary section.

I would be more careful with statements about phytoplankton preferences of DIP and DOP based on correlations of these variables, in addition to low nutrient ratios leading to lower phytoplankton biomass. Furthermore, I think that conclusion about a particular nutrient limitation based on the ratios might be misleading, unless there are clear indications of low nutrient concentrations. The ratio can be high, however the concentrations of bioavailable nutrients could be also high, thus none of the nutrients might be limiting phytoplankton growth. R: Agreed and thank you for pointing this out. The nutrient concentrations are presented in the results section. We will add a note on

the nutrient concentrations in the discussion and revise our argument regarding the correlation to phytoplankton.

Specific comments 1. 19-21 Rework the sentence "..for example, despite. . ." R: Done.

2. 25 "distribution fate?" R: The sentence was paraphrased to the following: Two sampling campaigns (August 2016, March 2017) were undertaken along ∼300 km of the Rajang river-South China Sea continuum to study both spatial and seasonal distribution of nutrients along the continuum.

3. 25-27 Place abbreviations of water quality samples in parenthesis R: Agreed, changed as recommended. It now reads: The analyses for nutrients encompass both inorganic i.e Nitrate (NO3-), Nitrite (NO2-), Ammonium (NH4+), Phosphate, DIP (PO4-) and Silicate, (dSi) as well as organic i.e dissolved organic nitrate (DON) and dissolved organic phosphate (DOP) fractions.

4. 30-32 It is hard to understand what is "removal" means. Did DIP decreased by 57.78%? R: Thank you for pointing this out. The term "removal" here refers to the terms utilized in the conservative index of mixing. The sentence now reads: "Both DIP and DOP exhibited non-conservative behaviour in the mixing according to the conservative index of mixing."

5. 32-33 Suggest rephrase, not clear: The bulk = major fraction of? R: Agreed, changed as recommended.

6. 33-35 Which preference is it? R: The term "preference" was changed with "stronger correlation". The sentence now reads: Spearman's correlations show that there was a stronger correlation of Chl a with DOP as compared to DIP when its concentrations are higher during the wet season.

7. 36-38 Back to general comments: what if the increased NO3:DIP ratio in wet season was due to higher discharge and consequently loading of NO3? How can ratios lead to anything? R: Thank you for this. It was assumed here that the wet season would result

in different loading of the different components. Based on the phytoplankton biomass, the lower biomass in the wet season justifies the assumption that it is probable that only the NO3 component increases, thus, increasing the ratio.

8. 65-67 Rephrase the sentence R: Agreed, this now reads: Due to the rapid economic development as a result of population growth, this results in the extensive modification tropical South East Asian rivers and degradation of catchments (Jennerjahn et al., 2008; Yule et al., 2010).

9. 76-79 Too convoluted sentence, suggest divide into 2 R: Agreed. The sentences now read: However, the Rajang river is tidal influenced and consists of fluvially-driven inputs of terrestrial mineral soils in the upper altitudes. It also drains peat domes in the lower altitudes (towards the coastal regions). Thus, it is imperative to understand the anthropogenic variability in nutrient dynamics in the landscape to better understand how such systems may respond to disturbance.

10. 81-83 Too convoluted, suggest divide into 2 R: Agreed, the sentences now read: A macronutrient that is essential but often limiting in freshwater systems is phosphorus (Elser et al., 2007). Under specific conditions, this macronutrient also limits the primary productivity of terrestrial and coastal ecosystems (Street et al., 2018; Sylvan et al., 2006).

11. 85-87 Rephrase, too wordy R: Agreed. The sentences now read: On a global scale, it was estimated that the riverine DIP loading for the world's largest rivers is 2.6 Tg yr-1 (Turner et al., 2002). These rivers represent 37% of the earth's watershed area and half of the earth's population.

12. 93-112 Talking about Carbon here, but that is not the focus of this study. Basically there is a need in a smoother transition between the gaps in knowledge and the goals of the study R: Thank you for highlighting this. We have modified this paragraph. The paragraph now reads: The disturbance of peatlands due to anthropogenic activities such as deforestation and conversion of peatlands for agricultural activities poses a

threat to the environment. As the carbon pools in tropical peatlands are globally significant, with the current estimates ranging from 40 to 90 Gt of C (Yu et al., 2010; Page et al., 2011; Warren et al., 2014), there is cause for concern. This is because such peat systems are typically ombrotrophic (i.e nutrient limited) whereby additions of nutrients from anthropogenic activities would lead to a significant increase in the oxidation of soil organic matter (Murdiyarso et al., 2010). This peat soil, when disturbed, changes from carbon sink into carbon source, contributing to the greenhouse gases in the atmosphere (Hirano et al., 2012; Hooijer et al., 2010). Recent studies of lateral transport of $CO_2$ in tropical peat-draining rivers (Müller et al., 2015; Wit et al., 2015) showed that the tropical peat-draining river of Maludam National Park seems to have a moderate amount of outgassing of $CO_2$ as compared to other peat-draining rivers globally. While the Rajang River is considered a medium-sized river based on its discharge (Sa'adi et al., 2017), 11% of its catchment area is part of the 15-19% global carbon peat pool in Southeast Asia (Page et al., 2011). Therefore, due to the knowledge gaps of tropical peat-draining rivers, particularly the Rajang River, it is essential to understand the influence of peat on the riverine nutrient (particulary phosphate) loading into the South China Sea. As the South China Sea supports one third of the global marine biodiversity (Ooi et al., 2013), the contribution of the Rajang River towards the South China Sea in terms of primary productivity cannot be ignored.

Therefore, the aim of this study is to 1) better understand the spatial and temporal distribution of nutrients in the Rajang river, with particular focus on dissolved inorganic phosphate (DIP) and dissolved organic phosphate (DOP) in the Rajang River with consideration to the peat-draining nature, diverse inputs and influences and 2) consequentially determine its influence on the phytoplankton biomass.

13. 118-120 Should go to figure caption. And similar paragraphs just occupy space and R: Agreed. The sentence "The red triangles represent the samples collected from the dry season whereas the blue circles represent the samples collected for the wet season." was removed.

[Figure]

14. 136-137 Change to "which can be thicker than 1 m" R: Agreed, changed as recommended.

15. 155-160 All these categories and classifications should be visualized on a study map (Fig. 1) R: The categories are rather difficult to visualize on a study map as it the main underlying classification is based on salinity does change according to seasons. We will try and create a supporting figure displaying the categories on a map.

16. 164-165 So in Methods it is stated that there were 2 surveys, while in Abstract 3 sampling campaigns (Line 23) are mentioned. How many were there? R: Thank you for pointing this out. There are 2 surveys. The value in the abstract was changed.

17. 194-201 Looks more like discussion R: Thank you for this. This was initially part of the discussion, but moved to the method section. We have moved it back to the discussion section.

18. 227 Change to "obtained from" R: Agreed, changed as recommended.

19. 248-254 Very confusing way of writing the equation. Why not state the equation, number it and explain the conversions, variables and units in the text? R: Agreed. The changes have been made in the corrected manuscript.

20. 260-262, 273-274 Again, should be in a figure caption, or removed. It is a strange way to start a paragraph/section R: Agreed, the sentences were removed.

21. 291 Starts with the same information as in 285. This should be cleaned R: Agreed, the sentences from 291 - 295 were removed.

22. 316-317 DIN:DIP would be definitely correlated with DIP, because there is DIP on both sides R: The term "correlated" was removed and replaced with "attributed". The sentence was to demonstrate that the high ratios were not due to the reduction in nitrogen but due to the lower overall DIP concentrations.

23. 320-321 Change or remove this sentence R: Agreed. The sentence now reads:

"Hence, for comparions and discussion in this study, the NO3-N:DIP were utilized instead of overall DIN:DIP."

24. 322-327 Which parameters are discussed? Was there any parameterization? R: The parameters are stated in the Method section under 2.5 Data Analyses. "For statistical correlations, SPSS (IBM SPSS Statistics 22) was utilized for calculations of independent sampling t-test (between seasons), one-way ANOVA (between source types) and Spearman's ranking (Bivariate correlation, for nutrients correlation)."

25. 350-351 DIP increases towards the sea while 358-359 says that there is a removal of DIP towards the coast. I am confused R: Thank you for highlighting this. The concentrations of DIP did increase towards the coastal region, however, in theory, the actual concentration should be higher than measured, as some of this DIP was removed along the river-sea continuum due to biogeochemical processes.

26. 374 Use "is" instead of "are" R: Agreed, changed as recommended.

27. 378-385 It is hard to understand the connection between the citation and the idea. I see that Funukawa et al 1996 stated that N and P are fairly high in soil solution, but how from this sentence 383- 383 can be concluded? R: Thank you for pointing this out. The sentence from 383 - 385 is an assumption that was inferred from the study done by Funakawa et al. whereby the loss of P during the rainy season was a result of run-off. We added the sentence "However, this inference requires further validation." as it is not a verified conclusion and removed the sentence of 378-381.

28. 394-395 Instead of "addition" it is better to use "increase" R: Thank you for pointing this out. However, the term "addition" here is specific as it refers to the conservative index of mixing. If the data falls on theoretical dilution line, no removal or addition occurs.

29. 411-412 How can DOP and DOC be compared? R: Thank you for this question. The DOC is used here as a proxy for peat which we then use to compare to the organic

portion of P.

30. 416-420 Back to general comments: the ratio can be high but the concentrations also can be high R: As mentioned above, we will add a note on the nutrient concentrations in the discussion.

31. 443-444 Chl a can be uncorrelated with DIP, but how is this reflected by NO3:DIP ratios? R: As mentioned above as well, we will revise our argument regarding the correlation to phytoplankton.

32. 447 Change "mass transport of biomass" R: Agreed, changed to "transport of biomass".

33. 473 Why use "Thus"? It is not clearly following from the previous sentence R: The word "thus" was removed.

34. 473-477 Is it really evident? I agree that DOP can be possibly utilized by phytoplankton, but the increase of DOP concentration does not indicate a preference switch. It is actually supported by discussion at 469-470 that DIP is easier to consume. R: Thank you for pointing this out. From the spearman's ranking, there was indeed a stronger correlation of Chl a with DOP as compared to DIP in the wet season. We have removed the sentence in question. It now reads: "As the Rajang River has a greater pool of DOP as compared to DIP (Fig.3.5(C)), the change of Chl a being positively correlated to DIP to DOP (Table 3) reflects a probable switch in the preference and utilization of DOP as compared to DIP as the preferred phosphate sources for the phytoplankton biomass."

35. 491-493 Still did not understand why the estimated figures are useful R: Thank you for this statement. Most of the work regarding global P estimations is based on models. Having an estimation based on actual concentrations of P in the river branches will hopefully aid to make P estimations more accurate.

36. 495-498 Too convoluted R: Agreed. The sentence now reads: Globally, it was

predicted that the river basins in both Central America and Southeast Asia (particularly Malaysia and Indonesia) would be hot spots (within the top 10% globally) for nutrient yields of various P forms (Seitzinger et al., 2005). However, based on calculations, the export of P from the Rajang River is comparatively minor when compared to other major rivers.

37. 498-499 It is unclear what exactly Seitzinger et al 2005 justifies R: The sentence has been rephrased to highlight the justification (discharge as a key driver of nutrient concentrations). The sentence now reads: The lower export of P from the Rajang river can be justified by Seitzinger et al., (2005), whereby the major driver that controls export of P and P forms is influenced by water discharge.

38. 505-508 Too convoluted, suggest split into at least 2 sentences R: Agreed, the sentence now reads: The comparison of dSi:DIP ratios to the yields of the Rajang showed that the DIP yields were variable and were likely due to anthropogenic sources. On the other hand, dSi originates from natural chemical and physical weathering, which are relatively stable compared to riverine N and P loads (Beusen et al., 2009).

39. 532-555 Needs additional work as Conclusion is largely based on the points mentioned above R: Agreed and revised based on corrections undertaken.

---

## Author Comment (AC2) · 4 Nov 2019

General comments

I would like to thank the authors for their obvious hard work on this manuscript. The role of phosphorus in driving primary productivity is a focus in many systems (Lake Erie, Lake Taihu, Gulf of Mexico, etc.) and so to understand how this relationship behaves in as many different systems as possible is fundamental to eventually being able to design control and remediation protocols. I am glad to see a study that looks at the different

fractions of dissolved phosphorus (DIP v. DOP), a subject that has been difficult to address in the past, but has been gaining in research focus recently. Additionally, I am pleased to see their focus on the role of the river itself as a fundamental actor in this relationship as opposed to an inert transporter of nutrients from one place to the next. The role of in-stream processing on nutrient loading is poorly understood, and by showing that there are real differences in nutrient concentrations along the entire length of the river helps to show that rivers are chemically dynamic systems. I believe that this study helps to progress the state of the science, and should be considered for publication after some changes, particularly to the grammar and sentence structure of the manuscript.

R: We would like express our gratitude to Ref #2 for the kind comments and acknowledging the work done for this manuscript. The comments and suggestions provided helped to improve the manuscript significantly.

General: As noted above, there are some issues with the language of the manuscript that makes it difficult to understand what the authors were trying to say. This become a problem in the discussion and conclusion sections where it seems the authors are contradicting themselves from one sentence to the next. I don't believe it is a misunderstanding or misinterpretation of their results, rather an issue with word choice and sentence structure.

R: We have employed a professional language editor to improve the language.

I think the comparisons with other rivers is a good idea, but ultimately executed poorly, it feels rushed and not properly fleshed out. I suggest that this section is a good starting point for another manuscript, but probably doesn't belong here.

R: We thank you for the suggestion and will prepare a second manuscript to flesh this out in the near future. For this manuscript, we would, however, prefer to keep our basic comparison with other river systems. The main reason is that the Rajang is the largest river in Malaysia and we feel it would be inappropriate to not compare it to any other

systems. Similarly, given the influence of the peat areas in the studied system, we do feel that it has to be mentioned and compared, even if only rudimentary.

Introduction: This section, in particular, will require editing/ rewriting. While the general structure of the section is fine, there are a significant number of grammatical issues which make reading and comprehension difficult. I have no issue with the message the authors are trying to convey; they did a fine job of providing supporting sources, however, it took several re-reads to be able to understand what they were trying to establish. Below are some examples of the confusing language used in this section, but is not a comprehensive list; these should be used as examples of what was outlined above.

R: As mentioned above, we have employed a professional language editor to improve the language of the overall manuscript.

Line 52: Awkward phrasing, try something like "The view of rivers as simply passive transporters of nutrients has been challenged in a number of recent studies (Richey et al., Tranvik et al.)

R: Agreed, the sentence now reads: "The view of rivers as passive transporters have recently been challenged by severa; studies (Richey et al., 2002; Tranvik et al., 2009)."

Line 59-60: Confusing wording- why nonetheless? The previous sentence sets up the fact that eutrophication is increasing.

R: Agreed. The word "nonetheless" was removed. The sentence now reads: "Eutrophication occurs due to enhanced nutrient levels which varies from one aquatic environment to another (Di and Cameron, 2002)."

Line 65-67: Sentence fragment. I think the authors are saying "The rapid increase in economic development, driven by population growth, has resulted in the modification of SE Asian rivers and the degradation of their catchments."

R: Agreed, the sentence now reads: "Due to the rapid economic development as a

result of population growth, it resulted in the extensive modification of tropical South East Asian rivers and degradation of catchments (Jennerjahn et al., 2008; Yule et al., 2010)."

There are numerous sentences like theses throughout the introduction, and they make the manuscript difficult to follow. The authors make some good points, and set up their study, it just takes a significant amount of effort to parse the language. This section has the ability to be a fine introduction if and when the language is corrected.

R: As mentioned above, we have employed a professional language editor to improve the language of the overall manuscript.

Methods: Study area: This section is fine, and the authors do a good job of describing their sampling locations/ decisions in selecting their sites. There are still some oddities in the language, but is ultimately easier to read and understand.

Sampling: Again, this section is generally fine, and does a good job of describing their sampling protocol, although I would ask how many samples were collected at each site as well as time of day for each collection. Are these single grab samples or are the authors averaging over a larger number of samples at each site? I may have missed it, but I did not see anything that describes this directly.

R: Thank you for this question. We have added the following sentence: "One sample was obtained for each site whereby a total of 29 samples were collected in the August 2016 campaign and a total of 16 samples were obtained in the March 2017 campaign."

Nutrient Analysis: I am not sold on the use of DIP as a proxy for PP, particularly in areas away from the estuaries, but I don't think it would have a significant impact on this study's results.

R: Noted.

Line 203: What fraction is it?

R: The word fraction was replaced with "portion". The sentence now reads: "In order to analyse correlation between humic acids and DIP or DOP, dissolved organic carbon concentrations (DOC) were used as a proxy for humic substances. This is because as the part of the hydrophobic fraction portion of dissolved organic matter (as DOC) forms part of humic substances are generally derived from humic substances (Findlay et al., 2003)."

CHL-a determination: The methods used are fine, although for blue-greens, chlorophyll can be misleading, and perhaps phycocyanin would be a better measure.

R: Thank you for the suggestion. As the chl a used here is just a proxy for phytoplankton biomass, the interpretation would remain the same. Furthermore, as the concentrations of chl a are already rather low, the concentrations for phycocyanin would be similarly low.

Data analysis: The methods outlined and statistical software used are fine. Export Calculations: I am not familiar with some of the calculations that they used here, but after looking into them I don't see anything that would raise any issues based on how they have described using them.

248-254: Is this the equation standard for this journal? Just seems like an odd way to write all of this out.

R: Thank you for pointing this out. The equations have been rectified in text.

Results: There are many of the same sort of language issues in this section that were present in the introduction. While it doesn't nullify the results it does make it difficult to read in a timely way. The results section is excessively wordy, and feels like it was written in several different pieces and then combined instead of being a singular effort.

R: Thank you for the comment. We have tried to summarise the results and employed a professional language editor to improve the overall language of the manuscript.

Lines 206-262: This seems like it should be a figure caption. There are a couple of

other spots in this section with the same sort of "disconnected" feel. If you can use figures or graphs, do so, and limit the amount of writing, particularly in a results section

R: Thank you for highlighting this. The sentences that form part of the figure captions were removed and appropriately placed as a figure caption.

Line 316 and elsewhere: Be careful in how you describe your DIN:DIP ratio comparisons. R: Noted and amended.

Discussion: I think that the language issues that came up in the introduction and results are present in this section as well. In a number of places it is not readily apparent what the authors are trying to say, and it takes multiple re-readings to understand. Additionally, there are a few places where they seem to contradict their own discussion points, but I think that it is through the use of incorrect phrasing as opposed to a misunderstanding of the results.

351-359: This is an example of what was described above, is it increasing or decreasing as it moves towards the coast?

R: Thank you for this question. The "removal" here refers to the theoretical conservative index of mixing. If the data falls on theoretical dilution line, no removal or addition occurs. The concentration is increasing, but at a lower rate than the theoretical amount.

378-381: What? Consider removing.

R: Thank you for this suggestion. We have removed the sentences as suggested.

416-418: Ratios are not concentrations are not loads. Flow weighting the loads could be helpful.

R: Thank you for the suggestion. We will calculate them and see if they aid the discussion.

485-529: I think that this is an important topic, but feels "jammed-in" here, and doesn't really advance the narrative in the way I think the authors wanted- if anything it muddies

things up a bit. I would cut this section way, way down or remove altogether. It is the seed of another manuscript to be honest and is not done justice here.

R: Thank you for this. We have reduced the content of this section and will prepare another manuscript in future.

Conclusion: Again, I think language issues hinder the author's ability of bring a significant amount of work to a fine enough point. The authors are trying to extend their results into places I'm not sure they actually go. This study is a good survey of the P exports of the river, and describes spatial and temporal variability in those measurements, but it is dangerous to compare to other systems (i.e. The Detroit River exports significantly more N and P to Lake Erie than the Maumee River, but the Maumee has an outsized role in harmful algal bloom formation due to the concentrations of those nutrients, and its relatively warmer water).

R: Thank you. As indicated above, we will follow your advice and prepare a second in-depth manuscript comparing various systems. As outlined above, we do feel that a short comparison to other river systems is necessary though. We have therefore reduced this section accordingly.

Tables and Figures: Table formatting is odd. This may be due to the way it printed out for me, but there are line jumps and returns that should be removed.

R: Thank you for pointing this out. The tables have been formatted to make it tidier.

Figure 2-4: the dots are difficult to see when printed out- mainly there is not enough contrast between the points and the map base layer.

R: Thank you for this. In the final version, we will include the high resolution pictures which will aid with the clarity and contrast of the figures.

Figure 5 and others: Be careful with axis font sizes, they are all over the place and make it difficult to read a number of the plots.

R: Thank you for highlighting this. The figures have been amended.

Figure 6 and 7: Look weirdly stretched out, like they were not resized properly.

R: Thank you for pointing this out. They have been rectified.